# Beyond Task Diversity: Provable Representation Transfer for Sequential Multi-Task Linear Bandits

**Thang Duong**
University of Arizona
thangduong@arizona.edu

**Zhi Wang**
University of Wisconsin–Madison
zhi.wang@wisc.edu

**Chicheng Zhang**
University of Arizona
chichengz@cs.arizona.edu

## Abstract

We study lifelong learning in linear bandits, where a learner interacts with a sequence of linear bandit tasks whose parameters lie in an $m$-dimensional subspace of $\mathbb{R}^d$, thereby sharing a low-rank representation. Current literature typically assumes that the tasks are *diverse*, i.e., their parameters uniformly span the $m$-dimensional subspace. This assumption allows the low-rank representation to be learned before all tasks are revealed, which can be unrealistic in real-world applications. In this work, we present the first nontrivial result for sequential multi-task linear bandits without the task diversity assumption. We develop an algorithm that efficiently learns and transfers low-rank representations. When facing $N$ tasks, each played over $\tau$ rounds, our algorithm achieves a regret guarantee of $\tilde{O}\left(Nm\sqrt{\tau} + N^{\frac{2}{3}}\tau^{\frac{2}{3}}dm^{\frac{1}{3}} + Nd^2 + \tau md\right)$ under the ellipsoid action set assumption. This result can significantly improve upon the baseline of $\tilde{O}\left(Nd\sqrt{\tau}\right)$ that does not leverage the low-rank structure when the number of tasks $N$ is sufficiently large and $m \ll d$. We also demonstrate empirically on synthetic data that our algorithm outperforms baseline algorithms, which rely on the task diversity assumption.

## 1 Introduction

Recommendation systems that interact with customers to promote the best items for each user have been widely adopted around the world. These interactions are often sequential and can be modelled as linear bandit problems [Abe and Long, 1999, Dani et al., 2008, Rusmevichientong and Tsitsiklis, 2010, Abbasi-Yadkori et al., 2011], where the characteristics of items can be represented as context vectors, and a user's preference for an item (i.e., reward) can be modelled using a linear combination of the context of the item. Even though the problem is typically high-dimensional, different users may exhibit similar preferences, leading to a low-dimensional underlying reward structure.

Motivated by this observation, there has been growing interest in representation learning within the context of linear bandits. For instance, in the item recommendation example, each session of interaction with a user can be seen as a linear bandit task, and similarity across tasks can be captured by the existence of a global feature extractor that applies to all problem instances.

Formally, we consider a problem where the learner sequentially faces $N$ $d$-dimensional linear bandit tasks, each with horizon $\tau$, with a key assumption that the reward predictors of the $N$ tasks, $\theta_1, \ldots, \theta_N$, lie in an $m$-dimensional linear subspace of $\mathbb{R}^d$. The goal of the learner is to minimize their meta (pseudo-)regret, which is the sum of regret across all tasks (see Equation (1) below), by exploiting the shared subspace structure.

38th Conference on Neural Information Processing Systems (NeurIPS 2024).

One naive approach is to solve each task independently using a base algorithm (such as Lin-UCB [Abbasi-Yadkori et al., 2011] or PEGE [Rusmevichientong and Tsitsiklis, 2010]); this approach, which we will henceforth refer to as the *individual single-task baseline*), would yield an upper bound on the meta-regret of $0\tilde{O}(Nd\sqrt{\tau})$. On the other hand, had the shared $m$-dimensional subspace been known beforehand, one would only need to estimate each reward predictor's projection onto the subspace; this leads to a meta-regret of $\tilde{O}(Nm\sqrt{\tau})$. In this work, we focus on the setting where $N$ and $\tau$ are large and $m \ll d$, the regime in which representation transfer learning would be beneficial.

Despite rich results for multi-task linear bandits in the parallel setting [e.g. Yang et al., 2020, Hu et al., 2021, Yang et al., 2022, Cella et al., 2023], progresses on multi-task bandits in the sequential setting have been relatively sparse. This can be attributed to the additional challenge of *meta-exploration*: in addition to exploration in each bandit learning task, one also needs to determine when (in which tasks) to acquire more information on the shared $m$-dimensional subspace representation. This is in contrast to the parallel setting, where algorithms that treat all tasks equally can achieve a near-optimal regret through a reduction to low-rank linear bandits [Hu et al., 2021, Jang et al., 2021a].

Under the assumption that the action sets are well-conditioned ellipsoids, Qin et al. [2022] design an efficient algorithm with a meta-regret of $\tilde{O}\left(Nm\sqrt{\tau} + dm\sqrt{\tau N}\right)$. However, it relies on an additional key assumption that the tasks are "diverse" in the $m$-dimensional subspace: more formally, for any subsequence of tasks $S$, the $m$-th eigenvalue of the task parameters' covariance matrix $\frac{1}{|S|} \sum_{n \in S} \theta_n \theta_n^\top$ is bounded away from zero (see Tripuraneni et al. [2021] for a related assumption in the supervised regression setting). However, this task diversity assumption is hard to verify and may not even hold in practice. Therefore, we raise the question:

*Can we design sequential multi-task bandit algorithms with provable low meta-regret without strong assumptions on task parameters, especially on task diversity?*

In this paper, we answer this question positively. Under mild assumptions that the action sets are well-conditioned ellipsoids and all task parameters have norms upper and lower bounded by constants, we design an algorithm with a meta-regret of $\tilde{O}\left(Nm\sqrt{\tau} + N^{\frac{2}{3}}\tau^{\frac{2}{3}}dm^{\frac{1}{3}} + Nd^2 + \tau md\right)$[1], providing the first nontrivial result for sequential multi-task linear bandit without the task diversity assumption. To the best of our knowledge, prior to our work, no regret bounds better than that of the individual single-task baseline (i.e., $o(Nd\sqrt{\tau})$) were known for this setting.

Our algorithm, BOSS, is based on a reduction to a bandit online subspace selection problem. Specifically, for each new task $n$, our algorithm chooses a subspace, represented by its canonical orthonormal basis $\hat{B}_n$, to approximate the ground-truth subspace $B$ and guide exploration. To address the challenge of meta-exploration, as choosing different $\hat{B}_n$'s leads to learning the projections of $\theta_n$ onto different subspaces/directions, our algorithm is designed to randomly meta-explore in some tasks and use them to learn $B$. Empirically, we demonstrate the effectiveness of our algorithm in a simulated adversarial environment where the task diversity assumption does not hold.

## 1.1 Related work

**Parallel representation transfer for multi-task linear bandit.** The parallel setting where the learner interacts with $N$ tasks simultaneously in each round was initially studied by Yang et al. [2020]. For the finite action setting, under some distributional assumptions on the action set in each round, they provide a total regret lower bound of $\Omega\left(N\sqrt{m\tau} + \sqrt{md\tau N}\right)$ and an algorithm that matches this up to logarithmic factors. For the infinite action setting, they provide a lower bound for the problem of $\Omega\left(Nm\sqrt{\tau} + d\sqrt{m\tau N}\right)$, which holds even under the task diversity assumption. Under the same task diversity assumption and in the infinite action set setting[2], Yang et al. [2022] present an algorithm with a regret guarantee of $\tilde{O}\left(Nm\sqrt{\tau} + d\sqrt{m\tau N}\right)$.

---

[1]$\tilde{O}$ hides $polylog(\tau, N, d)$ factor

[2]Although not mentioned explicitly, a careful examination of Yang et al. [2022] shows that its Lemma 4.2 uses the assumption that $\sigma_m\left((\theta_1, \ldots, \theta_N)\right) \geq \sqrt{\frac{N}{m}}$, which is a task diversity assumption.

| Algorithm | Prior info. | Task diversity | Regret guarantee(s) |
|---|---|---|---|
| Indep. PEGE for each task | None | No | $\tilde{O}\left(Nd\sqrt{\tau}\right)$ |
| Qin et al. [2022] | $m, \tau$ | Yes | $\tilde{O}\left(Nm\sqrt{\tau} + dm\sqrt{\tau N}\right), \Omega\left(Nm\sqrt{\tau} + d\sqrt{m\tau N}\right)$ |
| Bilaj et al. [2024] | None | Yes | at least $O\left(N\sqrt{\tau}(d-m)\log\left(1 + \frac{m}{\lambda_{min}^{\mathcal{A}}(d-m)}\right)\right)$ |
| BOSS (our work) | $N, m, \tau$ | No | $\tilde{O}\left(Nm\sqrt{\tau} + N^{\frac{2}{3}}\tau^{\frac{2}{3}}dm^{\frac{1}{3}} + Nd^2 + \tau md\right)$ |

Table 1: Comparisons of the settings, assumptions, and regret guarantees in this paper and previous works. A more comprehensive comparison is available in Table 2 of Appendix A.

For general action spaces, Hu et al. [2021] provide a regret guarantee of $\tilde{O}\left(N\sqrt{md\tau} + d\sqrt{\tau Nm}\right)$, albeit using a computationally inefficient algorithm. They also provide an extension for multi-task linear reinforcement learning under the assumption of low inherent Bellman error.

Unlike previous approaches, Cella et al. [2023] relaxes the requirement to know the subspace rank $m$ and the need for the task diversity assumption. Assuming that the action sets are finite and drawn from a specific distribution, by using trace-norm regularization for estimating task parameters and taking actions in a greedy fashion, they provide a regret upper bound of $\tilde{O}\left(N\sqrt{m\tau} + \sqrt{dm\tau N}\right)$ that matches the lower bound from Yang et al. [2020] up to logarithmic factors.

**Sequential representation transfer for multi-task linear bandit.** Compared with the parallel setting, the sequential setting is more challenging, where the learner only interacts with one task at a time; hence, even after many tasks, the reward predictors of the seen tasks may not span the underlying $m$-dimensional subspace. Qin et al. [2022] avoid this challenge by assuming a task diversity assumption, i.e., any large enough subset of tasks span the underlying $m$-dimensional subspace in a well-conditioned manner. With this, they provide a meta-regret guarantee $\tilde{O}\left(Nm\sqrt{\tau} + dm\sqrt{\tau N}\right)$ which nearly matches a lower bound of $\Omega\left(Nm\sqrt{\tau} + d\sqrt{m\tau N}\right)$. They also extend their analysis to a nonstationary representation setting where the global feature extractor can change over segments of tasks. Yang et al. [2022] study the sequential setting with an additional assumption that $\|\theta_n\| = 1$ for all tasks; however, it appears that there may be a non-trivial oversight in the analysis.[3]

Bilaj et al. [2024] study a related setting where the task parameters are i.i.d. sampled from a distribution with high variances from a $m$-dimensional subspace and with low variances in the orthogonal directions. They provide a meta-regret guarantee of at least $\tilde{O}\left(N\sqrt{\tau}(d-m)\log\left(1 + \frac{m}{\lambda_{min}^{\mathcal{A}}(d-m)}\right)\right)$, where $\lambda_{min}^{\mathcal{A}}$ is the smallest eigenvalue of the empirical covariance matrix of the actions taken. Since a linear bandit algorithm is expected to converge to pull the optimal arm at the end; $\lambda_{min}^{\mathcal{A}}$ may be constant when the action space is fixed. Thus, this bound can be as large as $\tilde{O}(Nd\sqrt{\tau})$. For a quick reference, see Table 1 for a comparison between our work and most related works. See also Appendix A for further discussions on related work.

## 2 Problem setup

We consider a sequence of linear bandit tasks of the same length $\tau$, described by the task parameters $\theta_1, \cdots, \theta_N \in \mathbb{R}^d$ chosen by an environment such that they satisfy Assumption 1. The learner solves $N$ tasks sequentially. In task $n$, for each time step $t = 1, \ldots, \tau$, the learner chooses an action $A_{n,t}$ from the action set $\mathcal{A}$ that satisfies Assumption 2 and receives a reward $r_{n,t} = A_{n,t}^{\top}\theta_n + \eta_{n,t}$, where

---

[3]In the proof of Theorem 5.1 therein, a key inequality, $\|\theta_n - \tilde{\theta}_n\|_2 \le \|\theta_n\|_2 - \|\tilde{\theta}_n\|_2$, was used (see Equation (21)); however, this inequality does not hold in general.

---

**Algorithm 1** Meta-Exploration procedure

---

1: **Input:** Task index $n$, exploration length $\tau_1$ (a multiple of $d$)
2: **for** $i \in [d]$ **do**
3:     Let $A_{n,t} = \lambda_0 e_i$ for $t = u(i-1) + 1, \cdots, ui$, where $u = \frac{\tau_1}{d}$
4: **end for**
5: **for** time step $t \leftarrow 1, \cdots, \tau_1$ **do**
6:     Take action $A_{n,t}$ and receive the reward $r_{n,t}$
7: **end for**
8: Compute $\hat{\theta}_n := \text{argmin}_{\theta \in \mathbb{R}^d} \frac{1}{\tau_1} \sum_{t=1}^{\tau_1} \left( \langle A_{n,t}, \theta \rangle - r_{n,t} \right)^2$
9: **for** time step $t \leftarrow \tau_1 + 1, \cdots, \tau$ **do**
10:     Take action $A_{n,t} \leftarrow \arg\max_{a \in \mathcal{A}} \left\langle a, \hat{\theta}_n \right\rangle$
11: **end for**

---

$\eta_{n,t}$ is independent, mean-zero 1-sub-Gaussian noise. The learner then moves on to task $n+1$ and repeats the same learning process.

**Assumption 1** (Low-rank representation). *Let $m < d$. For task parameters $\theta_1, \cdots, \theta_N$, there exist (i) a global feature extractor $B \in \mathbb{R}^{d \times m}$ with orthogonal columns and (ii) vectors $w_1, \ldots, w_N \in \mathbb{R}^m$, such that $\theta_i = B w_i$ for all $i \in [N]$.*

Given a semi-orthonormal matrix $U \in \mathbb{R}^{d \times m}$ (i.e., $U^\top U = I_m$), denote by $U_\perp \in \mathbb{R}^{d \times (d-m)}$ a matrix whose columns constitute an orthonormal basis of the orthogonal complement of $\text{span}(U)$, where we break ties in dictionary order[4]. We also denote the $i$-th column vector of $U$ by $U(i)$.

Following Rusmevichientong and Tsitsiklis [2010] and [Qin et al., 2022], we also assume that the action set $\mathcal{A}$ is an ellipsoid and the task parameters have bounded $\ell_2$ norms:

**Assumption 2** (Linear bandits with ellipsoid action sets). *The action set $\mathcal{A} := \left\{ x \in \mathbb{R}^d : x^\top M^{-1} x \leq 1 \right\}$ is an ellipsoid, where $M$ is a symmetric, positive definite matrix. In addition, there exist constants $\theta_{\min}$ and $\theta_{\max} \leq 1$ such that for all tasks $n \in [N]$, $\theta_{\min} \leq \|\theta_n\|_2 \leq \theta_{\max}$.*

We define the expected pseudo-regret for task $n$ as $R_\tau^n := \tau \cdot \max_{a \in \mathcal{A}} a^\top \theta_n - \mathbb{E}\left[ \sum_{t=1}^\tau A_{n,t}^\top \theta_n \right]$, and the meta-regret for all $N$ tasks as

$$R_\tau := \sum_{n=1}^N R_\tau^n = \sum_{n=1}^N \left[ \tau \cdot \max_{a \in \mathcal{A}} a^\top \theta_n - \mathbb{E}\left[ \sum_{t=1}^\tau A_{n,t}^\top \theta_n \right] \right]. \tag{1}$$

The learner's goal is to sequentially interact with each task in a way that minimizes its meta-regret.

## 3 Algorithm

Unlike in the parallel setting or the sequential setting with a task diversity assumption, here, the learner cannot directly learn the global feature extractor $B$. Instead, it needs to reason with the uncertainty about $B$ learned from the seen tasks.

**High-level idea of our approach.** To simultaneously learn $B$ online and utilize our (imperfect) knowledge of it, we solve the sequential multi-task bandit problem using a bi-level approach:

- At the lower level, for each task $n$, the learner has the option of invoking two base algorithms: one performs naive exploration that does not incorporate our knowledge on $B$, using a variant of full-dimensional linear bandit algorithm (PEGE [Rusmevichientong and Tsitsiklis, 2010], Algorithm 1); the other tries to incorporate a learned subspace $\hat{B}$ as prior knowledge to get reduced regret (Algorithm 2).

  Algorithm 1 and 2 can be viewed as performing meta-exploration and meta-exploitation respectively: Algorithm 1, while ignoring the low-dimension property of the tasks, produces unbiased estimators

---

[4]As we will see, other tie-breaking mechanisms would not change our algorithm and analysis; see Definition 9 and the discussion below in Appendix C for more details.

---

**Algorithm 2** Meta-Exploitation procedure

1: **Input:** Task index $n$, exploration length $\tau_2$ (a multiple of $m$), and the subspace orthonormal basis $\hat{B}_n \in \mathbb{R}^{d \times m}$
2: **for** $i \in [m]$ **do**
3:     Let $A_{n,t} = \lambda_0 \hat{B}_n(i)$ for $t = u(i-1) + 1, \cdots, ui$, where $u = \frac{\tau_2}{m}$
4: **end for**
5: **for** time step $t \leftarrow 1, \cdots, \tau_2$ **do**
6:     Take action $A_{n,t}$ and receive the reward $r_{n,t}$
7: **end for**
8: Compute $\hat{w}_n := \text{argmin}_{w \in \mathbb{R}^m} \frac{1}{\tau_2} \sum_{t=1}^{\tau_2} \left( \left\langle A_{n,t}, \hat{B}_n w \right\rangle - r_{n,t} \right)^2$
9: Let $\hat{\theta}_n := \hat{B}_n \hat{w}_n$
10: **for** time step $t \leftarrow \tau_2 + 1, \cdots, \tau$ **do**
11:     Take action $A_{n,t} \leftarrow \arg\max_{a \in \mathcal{A}} \left\langle a, \hat{\theta}_n \right\rangle$
12: **end for**

---

of $\theta_n$ that helps learn the task subspace $B$; Algorithm 2 allows the learner to achieve a much lower regret when the subspace $\hat{B}$ approximately contains $\theta_n$; however, using it may slow down the learning of $B$.

- At the upper level, the learner has two decisions to make for each task $n$: (1) choosing between meta-exploration and meta-exploitation; (2) choosing a subspace $\hat{B}_n$ to use if performing meta-exploitation. To this end, we propose Algorithm 3, which aims at making these decisions in a feedback-driven way to ensure low meta-regret.

We now elaborate on each level in more detail.

**The lower level.** As mentioned above, Algorithm 1 is for meta-exploration. When invoked in task $n$, it can achieve two goals simultaneously: obtaining an unbiased estimate of $\theta_n$, while maintaining a reasonable regret guarantee for task $n$. For the first $\tau_1$ steps (line 3 to 6), the learner takes actions $\{\lambda_0 e_i\}_{i=1}^d$ that span the action space $\mathcal{A}$, where $e_i$ is the $i$-th canonical vector of $\mathbb{R}^d$ and $\lambda_0 = \sqrt{\lambda_{\min}(M)}$ is a constant factor that ensures $\lambda_0 e_i \in \mathcal{A}$ (recall Assumption 2).

Then, the learner estimates task parameter $\hat{\theta}_n$ in line 8 and acts greedily for the rest of the task (line 10). We summarize its guarantee (originally due to Rusmevichientong and Tsitsiklis [2010]) as follows:

**Lemma 3.** *Fix $\tau_1$ to be a multiple of $d$. Suppose Algorithm 1 is run on task $n$ with the exploration length $\tau_1$. Then, there exists some constants $c_1, c_2 > 0$ (that depend on $\lambda_0, \theta_{\max}, \theta_{\min}$, and $M$) such that:*

*1. The regret on task $n$ is bounded as $R_\tau^n \leq c_1 \cdot \left( \tau_1 + \tau \cdot \frac{d^2}{\tau_1} \right) =: \boldsymbol{C_{info}}$;*

*2. With probability $\geq 1 - \delta$, $\|\hat{\theta}_n - \theta_n\| \leq c_2 \cdot \left( d \sqrt{\frac{\ln \frac{d}{\delta}}{\tau_1}} \right) =: \alpha$.*

We defer the proof of this lemma to Appendix D. Lemma 3 reveals a tradeoff between meta-exploration and regret minimization for task $n$: if $\tau_1$ is larger (say, closer to $\tau$), $\hat{\theta}_n$ estimates $\theta_n$ more accurately; however, this may yield a worse bound on $R_\tau^n$.

On the other hand, Algorithm 2 is for meta-exploitation. It takes a subspace (represented by its orthonormal basis $\hat{B}$) as input, and when invoked in task $n$, it can achieve a lower regret guarantee than Algorithm 1 when the subspace contains vectors that closely approximate $\theta_n$. Instead of exploring $\mathbb{R}^d$, the learner only explores the subspace induced by $\hat{B}_n$ (lines 3 and 6). Then, the learner estimates the low-dimensional task parameter $\hat{w}_n$ in line 9 and acts greedily for the rest of the task (line 11). We summarize its guarantee (originally due to Yang et al. [2020]) as follows:

**Lemma 4.** *Fix $\tau_2$ to be a multiple of $m$. Suppose Algorithm 2 is run on task $n$ with input subspace $\hat{B}_n$ and the exploration length $\tau_2$. Then, there exists some constant $c > 0$ (that depends on $\lambda_0, \theta_{\max}, \theta_{\min}$,*

**Algorithm 3** BOSS: Bandit Online Subspace Selection for Sequential Multitask Linear Bandits. The full algorithm 4 is in Appendix A.

---

1: **Input:** Task length $\tau$, number of task $N$, task dimension $d$, subspace dimension $m$, and exploration rate $p$.
2: **Initialize:** An uniform distribution $D_0$ over the set of experts $\mathcal{E}^\varepsilon$ in Definition 5
3: **for** $n \in [N]$: **do**
4:     Randomly draw $\hat{B}_n$ from $D_n$
5:     With probability $p$: $Z_n = 1$, otherwise $Z_n = 0$
6:     **if** $Z_n = 1$ **then**
7:         Exploration procedure in Algorithm 1
8:         Update the distribution $D_{n+1}$ with the EWA algorithm [Freund and Schapire, 1997], with learning rate $\eta = \ln 2$
9:     **else**
10:         Exploitation procedure in Algorithm 2 with $\hat{B}_n$
11:     **end if**
12: **end for**

---

*and $M$), such that the regret on task $n$ is bounded as:*

$$R_\tau^n \leq c \cdot \left( \tau_2 + \tau \cdot \left( \frac{m^2}{\tau_2} + \|\hat{B}_{n,\perp}^\top \theta_n\|_2^2 \right) \right).$$

*Specifically, if $\|\hat{B}_{n,\perp}^\top \theta_n\|_2 \leq 2\alpha$, then $R_\tau^n \leq 4c \left( \tau_2 + \tau \cdot \left( \frac{m^2}{\tau_2} + \alpha^2 \right) \right)$, where $\alpha$ is defined in Lemma 3.*

Lemma 4 reveals the opportunistic nature of Algorithm 2: if $\theta_n$ is perfectly contained in the subspace spanned by $\hat{B}_n$, $\|\hat{B}_{n,\perp}^\top \theta_n\| = 0$ and the regret bound is $R_\tau^n \leq O\left( \tau_2 + \tau \cdot \frac{m^2}{\tau_2} \right)$, which can be as low as $O(m\sqrt{\tau})$; on the other extreme, $\|\hat{B}_{n,\perp}^\top \theta_n\|$ can be as large as $\|\theta_n\|$ in the worst case, which yields a trivial linear regret bound. Thus, its regret guarantee hinges on good choices of subspace $\hat{B}_n$ as input. See Appendix E for the proof of Lemma 4.

**The upper level.** For the upper level, we propose Algorithm 3 that decides (1) when to perform meta-exploration and (2) the subspace to use if performing meta-exploitation.

For (1), for each task, the learner chooses to explore the subspace with probability $p$ (line 6) or exploit with the online subspace estimate $\hat{B}_n$ (line 9).

For (2), we propose to choose $\left\{ \hat{B}_n \right\}_{n=1}^N$ that can optimize the following cost function online[5]:

$$C_n(B) := \begin{cases} \mathbf{C}_{\text{hit}} := \tau_2 + \tau \cdot \left( \frac{m^2}{\tau_2} + \alpha^2 \right) & \|B_\perp^\top \theta_n\|_2 \leq 2\alpha; \\ \mathbf{C}_{\text{miss}} := \tau & \text{otherwise.} \end{cases} \tag{2}$$

The motivation behind this definition of $C_n$ is as follows: according to Lemma 4, $C_n(\hat{B}_n)$ is (up to constant) an upper bound of the regret of the learner at task $n$, were the learner to invoke Algorithm 2 using $\hat{B}_n$ for this task. Therefore, if we can guarantee $\sum_{n=1}^N C_n(\hat{B}_n)$ to be small, then using Algorithm 2 for all tasks yields a small meta-regret.

An immediate challenge in directly optimizing $C_n$ defined in Eq. (2) is its dependence on unobserved quantity $\theta_n$. This challenge is further complicated by the following: (i) at best, we observe $\hat{\theta}_n$'s that are $\alpha$-approximations of $\theta_n$ (e.g. in those meta-exploration tasks, see Lemma 3); (ii) in meta-exploitation tasks, we do not have guarantees on how close $\hat{\theta}_n$ is to $\theta_n$; note the difference in the definitions of $\hat{\theta}_n$ in meta-exploration and meta-exploitation tasks, respectively.

---

[5]In the case split of the definition of $C_n(B)$, the value of $\|B_\perp^\top \theta_n\|_2$ is independent of the choice of $B_\perp$. See Definition 9 and the discussion below in Appendix C for more details.

To address the challenge (i), we propose to optimize the following surrogate cost function for $C_n$:

$$\tilde{C}_n(B) := \begin{cases} \mathbf{C}_{\texttt{hit}} & \|B_\perp^\top \hat{\theta}_n\| \leq \alpha; \\ \mathbf{C}_{\texttt{miss}} & \text{otherwise.} \end{cases} \tag{3}$$

In Lemma 11 of Appendix C, we show that, with high probability, $\tilde{C}_n$ is an upper bound of $C_n$ (when $\hat{\theta}_n$ is close to $\theta_n$ as guaranteed by Lemma 3 in meta-exploration rounds).

With this modification of the optimization objective, challenge (ii) persists: $\tilde{C}_n$ is only a valid high-probability upper bound of $C_n$ for all meta-exploration rounds $n$; to ensure this upper bound property, we propose to optimize the cost:

$$\bar{C}_n(B) := \tilde{C}_n(B) \cdot \frac{Z_n}{p}, \tag{4}$$

where we introduce the importance weighting multiplier $\frac{Z_n}{p}$. Our key observation is that, although $\bar{C}_n(B)$ is no longer an upper bound of $C_n(B)$, the upper bound property holds *in-expectation*; to see this, observe that for any fixed $B$,

$$\mathbb{E}_{Z_n}\left[\bar{C}_n(B)\right] = \mathbb{E}_{Z_n}\left[\tilde{C}_n(B) \cdot \frac{Z_n}{p}\right] \gtrsim \mathbb{E}_{Z_n}\left[C_n(B) \cdot \frac{Z_n}{p}\right] = C_n(B),$$

here, the first equality is from the definition of $\bar{C}_n(B)$; the inequality (here, $\gtrsim$ indicates greater than up to a negligible constant) uses the above-mentioned property that when $Z_n = 1$, with high probability, $\tilde{C}_n(B) \geq C_n(B)$ (see Lemma 3 and Lemma 11); the second equality is from the fact that $Z_n \sim \text{Ber}(p)$.

Following the online learning literature, we propose to use the exponential weight algorithm (EWA, aka Hedge) [Freund and Schapire, 1997] to optimize $\{\bar{C}_n(B)\}$ online. Ideally, we would like to run EWA with the $\mathcal{B} = \{B : B \in \mathbb{R}^{d \times m} \text{ s.t. } B^\top B = I_m\}$, the set of all $m$-dimensional subspaces; however, this is impossible because $|\mathcal{B}|$ is infinite. So instead, we propose to run EWA with the expert set defined as an $\varepsilon$-cover of $\mathcal{B}$:

**Definition 5.** $\mathcal{E}^\varepsilon$ *is said to be a $\varepsilon$-cover over the set of $\mathcal{B}$ in the principal angle sense, if:*

$$\forall\, B \in \mathcal{B},\ \exists B' \in \mathcal{E}^\varepsilon \text{ such that } \|B_\perp^\top B'\|_{\mathrm{F}} \leq \varepsilon.$$

Definition 5 is motivated by the well-known fact that $\|B_\perp^\top B'\|_{\mathrm{F}}$ is the Frobenius norm of the sine of the principal angle matrix between subspaces spanned by $B$ and $B'$. In Appendix C.1 we show how to construct $\mathcal{E}^\varepsilon$ and its size $|\mathcal{E}^\varepsilon| \leq (\sqrt{dm}/\varepsilon)^{O(dm)}$. In subsequent discussions, we will assume that BOSS uses such an $\mathcal{E}^\varepsilon$.

Define a constant shift and scaling of $\bar{C}_n$, $\ell_n(B) := \frac{p}{\mathbf{C}_{\texttt{miss}}}\left[\bar{C}_n(B) - \mathbf{C}_{\texttt{hit}}\frac{Z_n}{p}\right]$; we note that any regret guarantee over sequence $\{\ell_n\}$ immediately translates to a regret guarantee over sequence $\{\bar{C}_n\}$. By the construction of the expert set $\mathcal{E}^\varepsilon$, sequence $\{\ell_n\}$ is realizable with high probability: there exists some $B_\varepsilon \in \mathcal{E}^\varepsilon$ such that $\sum_{n=1}^N \ell_n(B_\varepsilon) = 0$; this allows EWA to achieve a constant regret guarantee, summarized as follows:

**Lemma 6.** *Let $\varepsilon = \alpha = c_2 d\sqrt{\frac{\ln \frac{d}{\delta}}{\tau_1}}$ (with $c_2$ defined in Lemma 3) and $\delta = \frac{1}{N^2}$, where $c$ is a constant in Lemma 13. Then, assuming that $\tau \gg d^2$, Algorithm 3 chooses a sequence of subspaces $\left\{\hat{B}_n\right\}$ over the expert set $\mathcal{E}^\varepsilon$, defined in Definition 5, such that:*

$$\sum_{n=1}^N \mathbb{E}\left[C_n(\hat{B}_n)\right] \leq O\left(N\mathbf{C}_{\textit{hit}} + \frac{\mathbf{C}_{\textit{miss}} \log |\mathcal{E}^\varepsilon|}{p}\right) = \tilde{O}\left(N\left(\tau_2 + \tau \cdot \left(\frac{m^2}{\tau_2} + \alpha^2\right)\right) + \frac{\tau dm}{p}\right).$$

Note that the cumulative cost bound of $\left\{\hat{B}_n\right\}$ has two terms: the benchmark term $N\mathbf{C}_{\texttt{hit}}$ and the regret bound term $\mathbf{C}_{\texttt{miss}}\frac{\log |\mathcal{E}^\varepsilon|}{p}$. The benchmark term represents the best-case regret bound one can

achieve if, for all task $n$, the chosen subspace $\hat{B}_n$ can well approximate $\theta_n$ (i.e. $\|\hat{B}_{n,\perp}^\top \hat{\theta}_n\| \leq \alpha$) On the other hand, the regret-bound term depends on a few important factors: first, the cost decreases when the meta-exploration probability $p$ increases – this matches our intuition that a larger $p$ gives more frequent feedback to learn about $\theta_n$, allowing the learned $\hat{B}_n$ to adapt faster to $\theta_n$; second, the cost depends logarithmically on the size of the expert set $\ln |\mathcal{E}^\varepsilon|$, which is standard in the online learning from expert advice literature; third, the cumulative cost depends on the range of the instantaneous costs $\mathbf{C}_{\texttt{miss}}$.

**Remark 1.** The subspace selection game in the upper level resembles the partial monitoring problem [see Lattimore and Szepesvári, 2020, Chapter 37], where the learner does not directly observe the loss for its actions but receives signals from an environment according to an observation matrix. Here, in our subspace selection problem, the learner does not directly observe the chosen subspace's cost for most tasks, except when they choose to explore the subspace $B$. Unlike the traditional partial monitoring problem, here, the observation would be $\hat{\theta}_n$, and the cost $C_n(B)$ depends on the chosen subspace $B$ and $\theta_n$. This means that the cost matrix has an infinite number of columns (one for each $B$) and the observation depends on the actions of the learner and the environment in a stochastic fashion, unlike a deterministic dependence in the original partial monitoring setting.

## 4 Performance Guarantees

We bound the meta-regret of Algorithm 3 in Theorem 7:

**Theorem 7.** *With exploration probability* $p = \min\left(\left(\frac{2m\sqrt{\tau}}{N}\right)^{\frac{2}{3}}, 1\right)$, *by choosing* $\varepsilon = \alpha = c_2 d\sqrt{\frac{\ln \frac{d}{\delta}}{\tau_1}}$ *(with $c_2$ defined in Lemma 3)* , *where* $\delta = \frac{1}{N^2}$, $\tau_1 = d \cdot \left\lfloor \min\left(d\sqrt{\frac{\tau}{p}}, \tau\right)/d \right\rfloor$, $\tau_2 = m \cdot \lfloor \sqrt{\tau} \rfloor$, *the meta-regret of the* BOSS *algorithm is bounded by:*

$$R_\tau \leq \tilde{O}\left(Nm\sqrt{\tau} + N^{\frac{2}{3}}\tau^{\frac{2}{3}}dm^{\frac{1}{3}} + Nd^2 + \tau md\right). \tag{5}$$

In the meta-regret bound (5), we view the first two terms as the main terms and the last two as "burn-in" terms. The first term, $Nm\sqrt{\tau}$ is the cumulative regret bound of the oracle baseline, i.e. the idealized algorithm that takes advantage of the extra knowledge of $B$ to achieve a $O(m\sqrt{\tau})$ regret for every task. The second term, $N^{\frac{2}{3}}\tau^{\frac{2}{3}}dm^{\frac{1}{3}}$ is the main overhead for learning the representation $B$; it grows sublinearly in $N$, and as a consequence, is dominated by the first term when the number of tasks $N$ is very large (specifically, $N \gg \frac{d^3\sqrt{\tau}}{m^2}$). Compared with multi-task regret bounds in the parallel setting [Yang et al., 2020, Hu et al., 2021], our dependence on $N$ is admittedly weaker; nevertheless, to our knowledge, Theorem 7 is the first nontrivial result in the sequential setting without task diversity assumptions, which has not been studied before in [e.g., Qin et al., 2022].

For the burn-in terms, the $Nd^2$ term can be interpreted as a constant $d^2$ regret overhead per task; the $\tau md$ term can be interpreted as the learner sacrificing a constant number of tasks ($md$ tasks) to learn a good representation $B$. Observe that, in the less favourable situation where $N < md$ or $\tau < d^2$, the burn-in terms would lead to regret bounds worse than the trivial $N\tau$.

**Comparison with the individual single task baseline.** Recall that the individual single-task baseline has a meta-regret of $O(Nd\sqrt{\tau})$; our meta-regret guarantee improves over this baseline when $\tau \gg d^2$ and $N \gg m\sqrt{\tau}$. We leave broadening the parameter regimes when our guarantee outperforms the individual single-task baseline as an important open problem.

**Comparison with lower bounds.** Qin et al. [2022] showed a lower bound for the problem: $\Omega\left(Nm\sqrt{\tau} + d\sqrt{m\tau N}\right)$. We can see that there still exists a gap between our upper bound in Theorem 7 with this, and the gap is bigger than other solutions with task diversity assumption such as Qin et al. [2022]; we speculate that this is a price we pay due to not making any assumptions on task diversity.

We now sketch the proof of Theorem 7 below.

*Proof sketch.* Denote the pseudo-regret for task $n$ as $\hat{R}_\tau^n := \tau \max_{a \in \mathcal{A}} \langle \theta_n, a \rangle - \sum_{t=1}^\tau \langle \theta_n, A_{n,t} \rangle$.

We decompose $R_\tau$ as follows:

$$
\begin{aligned}
R_\tau &= \sum_{n=1}^{N} \mathbb{E}\left[\hat{R}_\tau^n\right] = \sum_{n=1}^{N} \mathbb{E}\left[\hat{R}_\tau^n Z_n + \hat{R}_\tau^n(1 - Z_n)\right] \\
&= \sum_{n=1}^{N} \mathbb{E}\left[\hat{R}_\tau^n \mid Z_n = 1\right] \cdot p + \sum_{n=1}^{N} \mathbb{E}\left[\hat{R}_\tau^n \mid Z_n = 0\right] \cdot (1 - p) \\
&\leq \mathbf{C}_{\texttt{info}} \cdot Np + \sum_{n=1}^{N} \mathbb{E}\left[C_n(\hat{B}_n)\right] \\
&= \tilde{O}\left(\left(\tau_1 + \tau \cdot \frac{d^2}{\tau_1}\right) Np + N\left(\tau_2 + \tau \cdot \left(\frac{m^2}{\tau_2} + \frac{d^2}{\tau_1}\right)\right) + \frac{\tau dm}{p}\right) \\
&= \tilde{O}\left(Np\tau_1 + N\tau\frac{d^2}{\tau_1} + \frac{\tau dm}{p} + N\tau_2 + N\tau\frac{m^2}{\tau_2}\right),
\end{aligned}
$$

where the first two equalities are by the definition of $R_\tau$ and algebra; the third equality uses the law of total expectation; the inequality uses Lemmas 3 and 4 to bound the first and second terms, respectively; the fourth equality is due to the definition of $\mathbf{C}_{\texttt{info}}$ and Lemma 6; the last equality is by algebra. □

The meta-regret of Algorithm 3 follows from the choices of $\tau_1, \tau_2$, and $p$ – specifically, $\tau_2$ balances the last two terms, whereas $\tau_1$ and $p$ aims at balancing the first three terms subject to the constraint that $\tau_1 \leq \tau$ and $p \leq 1$ – see Appendix G for the remaining details. □

**Adaptivity to problem parameters.** Algorithm 3 requires the knowledge of $N$ and $m$, the total number of tasks and the dimensionality of the subspace underlying the task parameters. Below, we show that knowledge of $N$ can be relaxed.

We can relax the need to know $N$ by using the doubling trick on BOSS. Specifically, in phase $i$, we can run our algorithm with the assumption that there are $2^i$ total tasks in this phase. The modified algorithm has a meta-regret guarantee that is within a constant of the algorithm that knows $N$. This implicitly gives an adaptive setting of meta-exploration probability $p$ that is decaying over time.

For $m$, the requirement can be relaxed to knowing an upper bound of $m$. Removing this knowledge requires a change of approach, such as low-rank matrix optimization, as in Cella et al. [2023] or additional assumption, as in Bilaj et al. [2024]. Cella et al. [2023] is in the parallel setting, which is not applicable here, and Bilaj et al. [2024]'s guarantee can be as large as $O(Nd\sqrt{\tau})$ as discussed in section 1.1.

## 5 Experiments

In this section[6], we compare the performance of our BOSS algorithm with the baselines on synthetic environments. The algorithms we evaluate include:

- PEGE: independently solves each task using the PEGE algorithm [Rusmevichientong and Tsitsiklis, 2010]
- PEGE-oracle: The "oracle baseline" that only uses PEGE on the true subspace $B$, for all tasks
- SeqRepL: our implementation of [Qin et al., 2022], in which $\hat{B}_n$ is estimated with SVD and the tasks for meta-exploration are chosen deterministically at round $n = \frac{i(i+1)}{2}$ for $i = 1, 2, \cdots$.
- BOSS-no-oracle: Algorithm 3 with $\mathcal{E}^\varepsilon$ set of 100,000 experts drawn uniformly at random from $\mathcal{B}$.
- BOSS: Algorithm 3 with $\mathcal{E}^\varepsilon$ set as 100,000 experts drawn uniformly at random from $\mathcal{B}$, plus the ground truth expert $B$. This algorithm is a better approximation of the original BOSS (Algorithm 3) since there exists $B_\varepsilon \in \mathcal{E}^\varepsilon$ such that $\|(B_\varepsilon)_\perp^\top B\|_F \leq \varepsilon$.

---

[6]The code for our paper can be found at https://github.com/duongnhatthang/BOSS

The setting is $(N, \tau, d, m) = (4000, 500, 10, 3)$. The environment reveals a new subspace dimension at tasks 1, 2501, and 3501. In this experiment, we assume $\mathcal{A}$ is a unit sphere, i.e., $M = I_d$. At each task $n$, denote by $B_n \in \mathbb{R}^{d \times m_n}$ the subspace basis that the environment used to generate $\theta_n$, where $m_n$ is incremented when $n = 1, 2501, 3501$. $\theta_n$ is chosen in the following way: $\theta_n = \lambda_1 B_n w_n$ for some $w_n \sim \text{Unif}(\mathbb{S}^{m_n - 1})$ and $\lambda_1 \sim \text{Unif}([0.8, 1])$ is a random scaling factor to ensures Assumption 2, where $\theta_{\min} = 0.8, \theta_{\max} = 1$.

The hyper-parameters $p, \tau_1, \tau_2$, and $\alpha$ of all algorithms, where it applies, are tuned. The error bands in the figures indicate $\pm 1$ standard deviation computed over 5 independent runs.

Figure 1a clearly shows the linear dependency of the cumulative regret on $N$. Observe that BOSS and its variants outperform both the independent PEGE and the SeqRepL baselines. It is also clear that the gap between BOSS-no-oracle and BOSS exists because the expert set $\mathcal{B}^{\varepsilon}$ used in this experiment does not cover the true $B$, since even with $\varepsilon = \frac{d}{\sqrt{\tau_1}} \approx 0.5$, the theoretical size of the expert set is $|\mathcal{E}^{\varepsilon}| = (\sqrt{dm}/\varepsilon)^{dm} \approx 11^{30}$ in this experiment setting which is much larger than the expert set size used in BOSS-no-oracle.

In Figure 1b, we plot $\|\hat{B}_{n,\perp}^{\top} B_n\|_{\mathrm{F}}$, which measures the closeness of $\hat{B}_{n,\perp}$ to $B_n$. When the environment reveals a new subspace dimension at tasks 1, 2501, and 3501, all algorithms' estimation $\hat{B}_n$ require updates and converge after a while. Even though BOSS-no-oracle has a worse estimation of $\hat{B}_n$ compared to SeqRepL, it achieves a better regret due to having a better estimation of $\hat{\theta}_n$ as shown in Figure 1c.

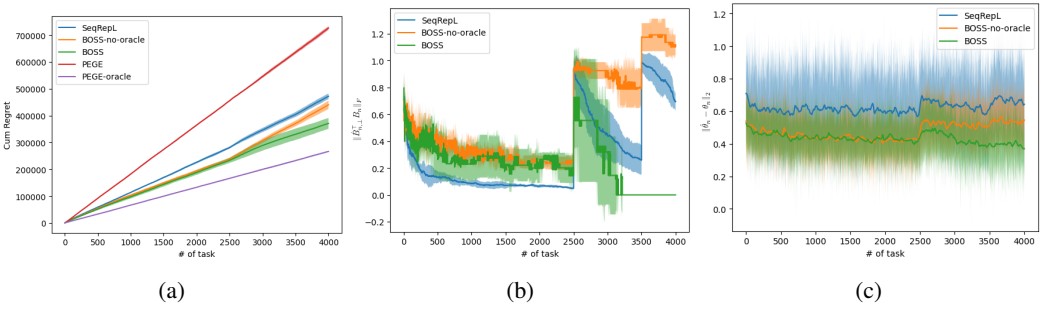

(a)  (b)  (c)

Figure 1: Comparing the cumulative regret of BOSS and other baselines. The setting is $(N, \tau, d, m) = (4000, 500, 10, 3)$ and $\|\theta_n\|_2 \in [0.8, 1] \; \forall n \in [N]$ chosen uniformly at random from this interval. The environment only reveals a new subspace dimension at tasks 1, 2501, and 3501, so there's no task diversity assumption.

# 6  Discussion and Future Work

We study the problem of sequential representation transfer in multi-task linear bandit, where the task parameters are allowed to be chosen adversarially online. Our BOSS algorithm achieves the regret guarantee of $\tilde{O}\left(Nm\sqrt{\tau} + N^{\frac{2}{3}}\tau^{\frac{2}{3}}dm^{\frac{1}{3}} + Nd^2 + \tau md\right)$ without using the task diversity assumption as in previous works. This opens up many promising avenues for future work. Statistically, it would be good to design an algorithm that performs no worse than the individual single-task baseline's performance in all parameter regimes. In addition, BOSS utilizes the special structure of fixed, ellipsoid-shaped action spaces to obtain useful information for meta-exploration, extending the algorithm and guarantees to general and time-varying action spaces is an important direction. Practically, it would also be nice to design parameter-free variants of BOSS that do not require knowing $m$ ahead of time. Furthermore, BOSS requires maintaining an exponentially large number of experts in $\mathcal{B}^{\varepsilon}$; in the future, we would like to develop more computationally-efficient algorithms. Lastly, it would be interesting to study relaxations of Assumption 1 (all task parameters lie exactly in a $m$-dimensional linear subspace), similar to Bilaj et al. [2024] or the changing subspace setting of Qin et al. [2022].

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

| Algorithm | Setting | Prior knowledge | Task diversity | Regret bound(s) |
|---|---|---|---|---|
| Indep. PEGE | Both | None | No | $\tilde{O}\left(Nd\sqrt{\tau}\right)$ |
| [Hu et al., 2021] | Parallel | $N, m$ | No | $\tilde{O}\left(N\sqrt{md\tau} + d\sqrt{\tau Nm}\right)$ |
| [Yang et al., 2020] | Parallel | $N, m, \tau$ | Yes | $O\left(Nm\sqrt{\tau} + d^{\frac{3}{2}}m\sqrt{\tau N}\right)$, $\Omega\left(Nm\sqrt{\tau} + d\sqrt{m\tau N}\right)$ |
| [Yang et al., 2022] | Parallel | $N, m, \tau$ | Yes | $\tilde{O}\left(Nm\sqrt{\tau} + d\sqrt{m\tau N}\right)$ |
| [Cella et al., 2023] | Parallel | $|\mathcal{A}_t| = K$ | No | $\tilde{O}\left(N\sqrt{md\tau} + d\sqrt{\tau Nm}\right)$, $\Omega\left(\sqrt{N\tau dm}\right)$ |
| [Jang et al., 2021b] | Sequential | None | No | $\tilde{O}\left(Nd\sqrt{\tau} + N^{\frac{3}{2}}\sqrt{d\tau}\right)$ |
| [Qin et al., 2022] | Sequential | $m, \tau$ | Yes | $\tilde{O}\left(Nm\sqrt{\tau} + dm\sqrt{\tau N}\right)$, $\Omega\left(Nm\sqrt{\tau} + d\sqrt{m\tau N}\right)$ |
| [Bilaj et al., 2024] | Sequential | None | Yes | at least $O\left(N\sqrt{\tau}(d-m)\right.$ $\left.\cdot \log\left(1 + \frac{m}{\lambda_{min}^{\mathcal{A}}(d-m)}\right)\right)$ |
| **BOSS (ours)** | Sequential | $N, m, \tau$ | No | $\tilde{O}\left(Nm\sqrt{\tau} + N^{\frac{2}{3}}\tau^{\frac{2}{3}}dm^{\frac{1}{3}} + Nd^2 + \tau md\right)$ |

Table 2: A comparison of the settings, assumptions, and regret guarantees between our results and prior works.

# A  Further discussion on related work

**Meta learning bandit problems.** Balcan et al. [2022] analyzed a harder problem in the Sequential setting by dealing with tasks generated by an adversary, and each task is an adversarial linear bandit problem. By using an online mirror decent base algorithm, they provided a guarantee of $\tilde{O}\left(\min_{\frac{1}{\tau} \leq \epsilon \leq \frac{1}{\sqrt{\tau}}} N\left(d\hat{V}_\epsilon\sqrt{\tau} + \epsilon\tau\right) + N^{\frac{3}{4}}\tau^2 d\right)$, where $\hat{V}_\epsilon$ measures the proximity of the optimal parameters of all $N$ tasks. Balcan et al. [2022]'s results also applies to the sequential meta-learning with adversarial $K$-arms bandit, which is further investigated by Azizi et al. [2024]. Their bandit meta-learning with a small set of optimal arms setting is analogous to the sequential multi-task representation transfer in linear bandit. The task diversity assumption is equivalent to the Azizi et al. [2024]'s Efficient Identification assumption; both require a gap to separate the noise from the problem's parameters. Our BOSS algorithm also has some high-level similarities with their E-BASS algorithm.

**Bilinear bandits.** The bilinear bandit problem described by Jang et al. [2021b] studies a setting where the learner has (potentially time-varying) sets of left actions and right actions available. It can take a left action $x_L$ and a right action $x_R$ and receive reward $r = x_L^\top \Theta x_R + \eta$. This reduces to our setting when the action set of the left action is the task descriptor $\{e_i\}_{i=1,\cdots,N}$ and the task's parameters have a low-rank structure. When applying their approach to our setting, their guarantee is $\tilde{O}\left(Nd\sqrt{\tau} + N^{\frac{3}{2}}\sqrt{d\tau}\right)$, which is worse than independently using a classical algorithm, such as PEGE, for each task. This is due to the fact that Jang et al. [2021b]'s solution does not exploit the low-rank and the left action structures.

In Table 2, we comprehensively compare the settings and assumptions of the previous works for the multi-task bandit representation transfer problem.

# B  Additional information on Algorithm 3

---

**Algorithm 4** BOSS: Bandit Online Subspace Selection for Sequential Multitask Linear Bandits

---

1: **Input:** Task length $\tau$, number of task $N$, task dimension $d$, subspace dimension $m$, learning rate $\eta$, and exploration rate $p$.
2: **Initialize:** An uniform distribution $D_1$ over the set of experts $\mathcal{E}^\varepsilon$ in Definition 5
3: **for** $n \in [N]$: **do**
4:      Randomly draw $\hat{B}_n$ from $D_n$
5:      With probability $p$: $Z_n = 1$, otherwise $Z_n = 0$
6:      **if** $Z_n = 1$ **then**
7:          Exploration procedure in Algorithm 1
8:          Update the distribution $D_{n+1}$ with the EWA algorithm
9:          For all $B \in \mathcal{E}^\varepsilon$, observe the cost $\tilde{C}_n(B) = \mathbb{I}(\|B_\perp^\top \hat{\theta}_n\|_2 \leq \alpha)\mathbf{C}_{\mathtt{hit}} + \mathbb{I}(\|B_\perp^\top \hat{\theta}_n\|_2 > \alpha)\mathbf{C}_{\mathtt{miss}}$
10:          The shifted and scaled loss:

$$\ell_n(B) = \frac{p}{\mathbf{C}_{\mathtt{miss}}}\left[\tilde{C}_n(B)\frac{Z_n}{p} - \mathbf{C}_{\mathtt{hit}}\frac{Z_n}{p}\right]\Bigg|_{Z_n=1} = \frac{1}{\mathbf{C}_{\mathtt{miss}}}\left[\tilde{C}_n(B) - \mathbf{C}_{\mathtt{hit}}\right]$$

11:          Update: $D_{n+1}(B) = \frac{D_n(B)\exp(-\eta\ell_n(B))}{\sum_{B'\in\mathcal{E}^\varepsilon} D_n(B')\exp(-\eta\ell_n(B'))}$
12:      **else**
13:          Exploitation procedure in Algorithm 2 with $\hat{B}_n$
14:          Update: $D_{n+1} = D_n$

$$\triangleright \text{ In this case, } \ell_n(B) = \frac{p}{\mathbf{C}_{\mathtt{miss}}}\left[\tilde{C}_n(B)\frac{Z_n}{p} - \mathbf{C}_{\mathtt{hit}}\frac{Z_n}{p}\right]\Bigg|_{Z_n=0} = 0$$

15:      **end if**
16: **end for**

---

# C  Additional details related to Section 3

We first provide more details on the construction of the expert set used in Section 3.

To construct $\mathcal{E}^\varepsilon$, an $\varepsilon$-cover of $\mathcal{B}$ in the principal angle sense (Definition 5) , we do the following:

1. Construct $\mathcal{E}_F^{\frac{\varepsilon}{2}}$, a proper $\frac{\varepsilon}{2}$-cover over $(\mathcal{B}_S, \|\cdot\|_F)$, where $\mathcal{B}_S$ is defined in Definition 8 (see below). Since $\mathcal{B} \subset \mathcal{B}_S$, $\mathcal{E}_F^{\frac{\varepsilon}{2}}$ is an also improper $\frac{\varepsilon}{2}$-cover over $(\mathcal{B}, \|\cdot\|_F)$. [7]

2. Construct $\mathcal{E}^\varepsilon$ from $\mathcal{E}_F^{\frac{\varepsilon}{2}}$ and show that it is a proper $\varepsilon$-cover over $(\mathcal{B}, \|\cdot\|_F)$

3. Show that $\mathcal{E}^\varepsilon$ is a proper $\varepsilon$-cover over $(\mathcal{B}, \|\cdot_\perp^\top\cdot\|_F)$.

We provide the details in Subsection C.1.

**Definition 8.** $\mathcal{E}_F^{\frac{\varepsilon}{2}}$ is said to be an $\frac{\varepsilon}{2}$-cover over the set of $\mathcal{B}_S = \{B : \|B\|_F \leq \sqrt{m}\}$ in the Frobenius norm sense, if:

$$\forall\ B \in \mathcal{B}_S,\ \exists A \in \mathcal{E}_F^{\frac{\varepsilon}{2}} \text{ such that } \|\mathrm{vec}(A) - \mathrm{vec}(B)\|_2 \leq \frac{\varepsilon}{2}$$

---

[7]We follow the convention in Telgarsky [2021] that a proper cover has to be a subset of the ground set, while an improper cover may not satisfy such property.

After defining the expert set, to use the EWA algorithm, we need to use a surrogate expert's cost $\tilde{C}_n$ defined in Equation (3), which is a high probability upper-bound of the true cost $C_n(B)$ following Lemma 11. To construct $\tilde{C}_n$, we need to introduce the definition of $\alpha$-covering in Definition 9.

**Definition 9.** *A subspace (represented by its orthonormal basis) $B \in \mathbb{R}^{d \times m}$ is said to $\alpha$-approximately cover vector $\phi \in \mathbb{R}^d$, if $\|B_\perp^\top \phi\|_2 \leq \alpha$.*

**Note:** the specific choice of $B_\perp$ does not affect the validity of this definition – e.g., $\|B_\perp \phi\|_2$ are always the same regardless of the specific choice of $B_\perp$, as long as its columns form a orthonormal basis of $\mathrm{span}(B_\perp)$: for any two choices of $B_\perp$ (denoted by $B_{\perp,1}$, $B_{\perp,2}$, respectively) there exists orthonormal $V \in \mathbb{R}^{(d-m) \times (d-m)}$ such that $B_{\perp,1} = B_{\perp,2}V$. Therefore,

$$\|B_{\perp,1}^\top \theta_n\|_2 = \|V^\top B_{\perp,2}^\top \theta_n\|_2 = \|B_{\perp,2}^\top \theta_n\|_2$$

The following lemma justifies that all valid choices of $B_\perp$ are equivalent up to a $(d-m) \times (d-m)$ orthogonal transformation:

**Lemma 10.** *Let $W$ be a $k$-dimensional subspace of $\mathbb{R}^d$. Let $B, \hat{B} \in \mathbb{R}^{d \times k}$ be matrices whose columns form an orthonormal basis of $W$. Then, there exists an orthogonal matrix $V \in \mathbb{R}^{k \times k}$ such that $\hat{B} = BV$.*

*Proof.* Since $B$ is a basis of $W$, there exists some $V$ such that $\hat{B} = BV$. Since $V^\top V = V^\top (B^\top B)V = (BV)^\top (BV) = \hat{B}^\top \hat{B} = I$, $V$ is an orthogonal matrix. $\square$

Next, we justify the use of $\tilde{C}_n$ as a surrogate cost for the true cost $C_n$ in Lemma 11, which requires Remark 2.

**Remark 2.** By the observation that

$$\|B_\perp^\top \phi\|_2 = \|B_\perp B_\perp^\top \phi\|_2 = \|\phi - BB^\top \phi\|_2 = \min_{\theta \in \mathrm{span}(B)} \|\phi - \theta\|,$$

$B$ $\alpha$-approximately covers $\phi$ if and only if there exists some $\theta$ in $\mathrm{span}(B)$ that is $\alpha$-close to $\phi$. As a result:

- If $\theta_n \in \mathrm{span}(B)$ and $\|\theta_n - \hat{\theta}_n\| \leq \alpha$, $B$ also $\alpha$-covers $\hat{\theta}_n$;

- If $B$ $\alpha$-covers $\hat{\theta}_n$ and $\|\theta_n - \hat{\theta}_n\| \leq \alpha$, $B$ also $2\alpha$-covers $\theta_n$.

**Lemma 11.** *If $\|\hat{\theta}_n - \theta_n\| \leq \alpha$, $\tilde{C}_n$ is an upper bound of $C_n$. Equivalently,*

$$\mathbb{I}(\|\hat{\theta}_n - \theta_n\| \leq \alpha)C_n(B) \leq \mathbb{I}(\|\hat{\theta}_n - \theta_n\| \leq \alpha)\tilde{C}_n(B)$$

*Proof.* Recall that

$$C_n(B) = \begin{cases} \mathbf{C}_{\mathtt{hit}}, & \|B_\perp^\top \theta_n\| \leq 2\alpha \\ \mathbf{C}_{\mathtt{miss}}, & \|B_\perp^\top \theta_n\| > 2\alpha \end{cases}, \quad \tilde{C}_n(B) = \begin{cases} \mathbf{C}_{\mathtt{hit}}, & \|B_\perp^\top \theta_n\| \leq 2\alpha \\ \mathbf{C}_{\mathtt{miss}}, & \|B_\perp^\top \theta_n\| > 2\alpha \end{cases},$$

First, observe that $\tilde{C}_n(B) \geq \mathbf{C}_{\mathtt{hit}}$ for any $B$. Then, we conduct a case analysis:

- Case 1: $\|B_\perp^\top \theta_n\|_2 \leq 2\alpha$. In this case, $C_n(B) = \mathbf{C}_{\mathtt{hit}} \leq \tilde{C}_n(B)$.

- Case 2: $\|B_\perp^\top \theta_n\|_2 > 2\alpha$. In this case, $C_n(B) = \mathbf{C}_{\mathtt{miss}}$. We now show that $\tilde{C}_n(B) = \mathbf{C}_{\mathtt{miss}}$. Indeed,

$$\|B_\perp^\top \theta_n\|_2 \leq \|B_\perp^\top \hat{\theta}_n\|_2 + \|B_\perp^\top (\hat{\theta}_n - \theta_n)\|_2$$
$$\leq \|B_\perp^\top \hat{\theta}_n\|_2 + \alpha$$
$$\implies 2\alpha < \|B_\perp^\top \hat{\theta}_n\|_2 + \alpha \qquad (\|B_\perp^\top \theta_n\|_2 > 2\alpha)$$
$$\implies \alpha < \|B_\perp^\top \hat{\theta}_n\|_2$$

where the first inequality is by triangle inequality; the second inequality is by the definition of this case and that $\|B_\perp^\top (\hat{\theta}_n - \theta_n)\|_2 \leq \|\hat{\theta}_n - \theta_n\| \leq \alpha$.

In summary, in both cases, $C_n(B) \leq \tilde{C}_n(B)$. □

To ensure that there exists some $B \in \mathcal{E}^\varepsilon$ that can $\alpha$-cover all $\hat{\theta}_n$, we need to choose $\varepsilon$ by following Lemma 12.

**Lemma 12.** *By choosing $\varepsilon \leq \alpha$, there exists some $B_\varepsilon \in \mathcal{E}^\varepsilon$ such that, for all $n$, $B_\varepsilon$ $\alpha$-approximately covers $\theta_n$.*

*Proof.* For any $n \in [N]$, $\theta_n \in \mathrm{span}(B)$. Hence, $\theta_n = P_B \theta_n$.

Thus,

$$
\begin{aligned}
\min_{U \in \mathcal{E}^\varepsilon} \|\theta_n - U U^\top \theta_n\|_2 &= \min_{U \in \mathcal{E}^\varepsilon} \|U_\perp U_\perp^\top \theta_n\|_2 \\
&= \min_{U \in \mathcal{E}^\varepsilon} \|U_\perp U_\perp^\top B B^\top \theta_n\|_2 \\
&\leq \min_{U \in \mathcal{E}^\varepsilon} \|U_\perp\|_{op} \|U_\perp^\top B\|_{\mathrm{F}} \|B^\top\|_{op}^2 \|\theta_n\|_2 \\
&\leq \min_{U \in \mathcal{E}^\varepsilon} \|U_\perp^\top B\|_{\mathrm{F}} \theta_{\max} \\
&\leq \varepsilon \theta_{\max}.
\end{aligned}
$$

Hence, assuming that $\theta_{\max} \leq 1$, by setting $\varepsilon \leq \alpha$, we have that $\min_{U \in \mathcal{E}^\varepsilon} \|\phi - U U^\top \phi\|_2 \leq \alpha$. □

### C.1 The expert set $\mathcal{E}^\varepsilon$: construction and size

First, we construct the surrogate set $\mathcal{E}_F^{\frac{\varepsilon}{2}}$, a proper $\frac{\varepsilon}{2}$-cover over $(\mathcal{B}_S, \|\cdot\|_F)$. Following Erdogdu and Vural [2024], we construct it by discretizing the hyper-cube that contains the ball $\mathcal{E}_F^{\frac{\varepsilon}{2}}$. Then $|\mathcal{E}_F^{\frac{\varepsilon}{2}}| \leq \left(\frac{4m\sqrt{d}}{\varepsilon}\right)^{dm} = (\frac{dm}{\varepsilon})^{O(dm)}$. Note that for any $B \in \mathcal{B}$, $\|B\|_F = \|\mathrm{vec}(B)\|_2 = \sqrt{m}$; therefore, $\mathcal{B} \subset \mathcal{B}_S$, and thus $\mathcal{E}_F^{\frac{\varepsilon}{2}}$ is an improper $\frac{\varepsilon}{2}$-cover of $\mathcal{B}$.

Then, we follow a procedure similar to [Telgarsky, 2021, Remark 15.3] to construct $\mathcal{E}^\epsilon$ as follows:

$$
\mathcal{E}^\epsilon = \left\{ U(b) : b \in \mathcal{E}_F^{\frac{\varepsilon}{2}} \right\},
$$

where $U(b) := \mathrm{argmin}_{B \in \mathcal{B}} \|B - b\|_F$. In words, $\mathcal{E}^\epsilon$ is the collections of "nearest neighbors" of $\mathcal{E}^\epsilon$ in $\mathcal{B}$. By the definition of $\mathcal{E}^\epsilon$, $\mathcal{E}^\epsilon \subset \mathcal{B}$ and $|\mathcal{E}^\epsilon| \leq |\mathcal{E}_F^{\frac{\varepsilon}{2}}| \leq (\frac{dm}{\epsilon})^{O(dm)}$.

For the rest of the subsection, we will show that our construction of $\mathcal{E}^\epsilon$ satisfies Definition 5.

**Lemma 13.**

1. *$\mathcal{E}^\varepsilon$ is a proper $\varepsilon$-cover of $(\mathcal{B}, (A, B) \mapsto \|A - B\|_F)$.*

2. *$\mathcal{E}^\varepsilon$ is a proper $\varepsilon$-cover of $(\mathcal{B}, (A, B) \mapsto \|A_\perp^\top B\|_F)$.*

*Proof.* For the first item, we consider any $B \in \mathcal{B}$. We would like to show that there exists some element $C$ in $\mathcal{E}^\epsilon$ such that $\|C - B\|_F \leq \varepsilon$.

We find element $C$ as follows: first, by definition of $\mathcal{E}_F^{\frac{\varepsilon}{2}}$, there exists element $b \in \mathcal{E}_F^{\frac{\varepsilon}{2}}$ such that $\|B - b\|_F \leq \frac{\varepsilon}{2}$. We define $C = U(b)$. Therefore, $\|C - b\| \leq \|B - b\|_F \leq \frac{\varepsilon}{2}$. Thus:

$$
\|B - C\|_F \leq \|B - b\|_F + \|C - b\|_F \leq \frac{\varepsilon}{2} + \frac{\varepsilon}{2} \leq \varepsilon.
$$

Hence, $\mathcal{E}^\varepsilon$ is a proper $\varepsilon$-cover over $(\mathcal{B}, \|\cdot\|_F)$.

We now prove the second item. It suffices to show that, for $A, B \in \mathcal{B}$, $\|A_\perp^\top B\|_F^2 \leq \|A - B\|_F^2$.

Following [Vu, 2020], for semi-orthogonal matrices $A, B \in \mathcal{B}$, we relate $\|A_\perp^\top B\|_F$ to the orthogonal Procrustes problem in [Gower and Dijksterhuis, 2004],

$$
\min_{R \in \mathbb{R}^{m \times m} : R^\top R = I_m} \|AR - B\|_F^2.
$$

With the solution $R^* = A^\top B \left(B^\top AA^\top B\right)^{-\frac{1}{2}}$ at the optimum,

$$\|A_\perp^\top B\|_{\mathrm{F}}^2 = \sum_{i=1}^m \sin^2(\phi_i) \qquad\qquad (\phi \text{ is the canonical angle})$$

$$= m - \sum_{i=1}^m \cos^2(\phi_i)$$

$$\leq m - \sum_{i=1}^m \left(2\cos(\phi_i) - 1\right) \qquad\qquad (\cos(\phi_i)^2 \geq 2\cos(\phi_i) - 1)$$

$$= 2m - 2\sum_{i=1}^m \cos(\phi_i)$$

$$= \|AR^* - B\|_{\mathrm{F}}^2. \qquad\qquad (\text{See [Vu, 2020]})$$

We have $\|A_\perp^\top B\|_{\mathrm{F}}^2 \leq \|AR^* - B\|_{\mathrm{F}}^2 \leq \|A - B\|_{\mathrm{F}}^2$.

$\qquad\qquad\qquad\qquad\qquad\qquad\qquad\qquad\qquad\qquad\qquad\qquad\qquad\qquad\qquad\qquad\qquad\quad\square$

## D   Proof of Lemma 3: Regret of Meta-exploration Tasks

We first restate Lemma 3.

**Lemma 3.** *Fix $\tau_1$ to be a multiple of $d$. Suppose Algorithm 1 is run on task $n$ with the exploration length $\tau_1$. Then, there exists some constants $c_1, c_2 > 0$ (that depend on $\lambda_0, \theta_{\max}, \theta_{\min}$, and $M$) such that:*

*1. The regret on task $n$ is bounded as $R_\tau^n \leq c_1 \cdot \left(\tau_1 + \tau \cdot \frac{d^2}{\tau_1}\right) =: \boldsymbol{C_{info}}$;*

*2. With probability $\geq 1 - \delta$, $\|\hat\theta_n - \theta_n\| \leq c_2 \cdot \left(d\sqrt{\frac{\ln\frac{d}{\delta}}{\tau_1}}\right) =: \alpha$.*

*Proof.* For the first item, following [Rusmevichientong and Tsitsiklis, 2010, Lemma 3.4], we have

$$\mathbb{E}\left[\|\hat\theta_n - \theta_n\|^2\right] \leq c_0 \frac{d^2}{\tau_1},$$

where $c_0$ is a constant that depends on $\lambda_0$ and $M$. By [Yang et al., 2020, Lemma 17], we have that $\max_{a\in\mathcal{A}}\langle a - A_{n,t}, \theta_n\rangle \leq J\|\theta_n - \hat\theta_n\|^2/\|\theta_n\|$, where $A_{n,t} = \mathrm{argmax}_{a\in\mathcal{A}}\left\langle a, \hat\theta_n\right\rangle$ and $J = \frac{\lambda_{max}(M)}{\sqrt{\lambda_{min}(M)}} = \frac{\lambda_{max}(M)}{\lambda_0}$. Thus,

$$R_\tau^n = \mathbb{E}\left[\tau \max_{a\in\mathcal{A}}\langle\theta_n, a\rangle - \sum_{t=1}^\tau \langle\theta_n, A_{n,t}\rangle\right]$$

$$= \mathbb{E}\left[\tau_1 \max_{a\in\mathcal{A}}\langle\theta_n, a\rangle - \sum_{t=1}^{\tau_1}\langle\theta_n, A_{n,t}\rangle + (\tau - \tau_1)\max_{a\in\mathcal{A}}\langle\theta_n, a\rangle - \sum_{t=\tau_1+1}^\tau \langle\theta_n, A_{n,t}\rangle\right]$$

$$\leq \lambda_0\theta_{\max}\tau_1 + (\tau - \tau_1)J\frac{\mathbb{E}\|\hat\theta_n - \theta_n\|^2}{\theta_{\min}}$$

$$\leq \lambda_0\theta_{\max}\tau_1 + \frac{J}{\theta_{\min}}\cdot \tau c_0\frac{d^2}{\tau_1}.$$

The proof of the first item is concluded by taking $c_1 = \max(\lambda_0\theta_{\max}, \frac{Jc_0}{\theta_{\min}})$.

For the second item, recall that $u = \frac{\tau_1}{d}$, and $A_{n,1}, \ldots, A_{n,\tau_1}$ are constructed as follows: for each $i \in [d]$ and $t \in \{u(i-1) + 1, \cdots, ui\}$, $A_{n,t} = \lambda_0 e_i$.

Since $\hat{\theta}_n := \arg\min_\theta \frac{1}{\tau_1} \sum_{t=1}^{\tau_1} (\langle A_{n,t}, \theta \rangle - r_{n,t})^2$, by the closed-form solution of ordinary least squares, we have

$$\hat{\theta}_n = (A_n^\top A_n)^{-1} A_n^\top r_n,$$

where

$$A_n := \begin{pmatrix} A_{n,1}^\top \\ \cdots \\ A_{n,\tau_1}^\top \end{pmatrix} \in \mathbb{R}^{\tau_1 \times d},$$

and $r_n := (r_{n,1}, \cdots, r_{n,\tau_1})$. Observe that

$$A_n^\top A_n = u\lambda_0^2 \sum_{i=1}^d e_i e_i^\top = u\lambda_0^2 I_d. \tag{6}$$

Let $\eta_n := (\eta_{n,1}, \ldots, \eta_{n,\tau_1})$. We have

$$\begin{aligned}
\hat{\theta}_n &= (A_n^\top A_n)^{-1} A_n^\top r_n \\
&\overset{(a)}{=} \frac{1}{u\lambda_0^2} A_n^\top r_n \\
&= \frac{1}{u\lambda_0^2} A_n^\top (A_n \theta_n + \eta_n) \\
&\overset{(b)}{=} \theta_n + \frac{1}{u\lambda_0^2} A_n^\top \eta_n,
\end{aligned}$$

where both (a) and (b) follow from Eq. (6).

It now suffices to show that $\left\| \hat{\theta}_n - \theta_n \right\|_2 = \left\| \frac{1}{u\lambda_0^2} A_n^\top \eta_n \right\|_2 \le O\left( d\sqrt{\frac{\log(d/\delta)}{\tau_1}} \right)$ with probability at least $1 - \delta$. To this end, observe that by the construction of $A_n$,

$$A_n^\top \eta_n = \begin{pmatrix} \lambda_0 \sum_{t=1}^u \eta_{n,t} \\ \cdots \\ \lambda_0 \sum_{t=\tau_1-u+1}^{\tau_1} \eta_{n,t} \end{pmatrix}.$$

Since for each $n$ and $t$, $\eta_{n,t}$ is zero-mean and 1-sub-Gaussian, by [Lattimore and Szepesvári, 2020, Corollary 5.5], for any $i \in [d]$, we have

$$\Pr\left( \left| \sum_{t=u(i-1)+1}^{ui} \eta_{n,t} \right| \ge \sqrt{2u \log(2/\delta')} \right) \le \delta'.$$

Let $\delta = d\delta'$ and

$$F_n := \left\{ \forall i \in [d], \quad \left| (A_n^\top \eta_n)_i \right| \le \lambda_0 \sqrt{2u \log(2d/\delta)} \right\}.$$

Then, by the union bound, $F_n$ happens with probability at least $1 - \delta$. Under the event $F_n$,

$$\begin{aligned}
\|\hat{\theta}_n - \theta_n\|_2 &= \left\| \frac{1}{u\lambda_0^2} A_n^\top \eta_n \right\|_2 \\
&= \frac{1}{u\lambda_0^2} \sqrt{\sum_{i=1}^d (A_n^\top \eta_n)_i^2} \\
&\le \frac{1}{u\lambda_0} \sqrt{d \cdot 2u \log(2d/\delta)} \\
&\le O\left( d\sqrt{\frac{\log(d/\delta)}{\tau_1}} \right),
\end{aligned}$$

and the proof of the second item is complete. $\qquad\square$

# E  Proof of Lemma 4: Regret of Meta-exploitation Tasks

We first restate Lemma 4.

**Lemma 4.** *Fix $\tau_2$ to be a multiple of $m$. Suppose Algorithm 2 is run on task $n$ with input subspace $\hat{B}_n$ and the exploration length $\tau_2$. Then, there exists some constant $c > 0$ (that depends on $\lambda_0, \theta_{\max}, \theta_{\min}$, and $M$), such that the regret on task $n$ is bounded as:*

$$R_\tau^n \le c \cdot \left( \tau_2 + \tau \cdot \left( \frac{m^2}{\tau_2} + \|\hat{B}_{n,\perp}^\top \theta_n\|_2^2 \right) \right).$$

*Specifically, if $\|\hat{B}_{n,\perp}^\top \theta_n\|_2 \le 2\alpha$, then $R_\tau^n \le 4c \left( \tau_2 + \tau \cdot \left( \frac{m^2}{\tau_2} + \alpha^2 \right) \right)$, where $\alpha$ is defined in Lemma 3.*

*Proof.* Similar to the approach in [Rusmevichientong and Tsitsiklis, 2010], define $J = \frac{\lambda_{max}(M)}{\sqrt{\lambda_{min}(M)}} = \frac{\lambda_{max}(M)}{\lambda_0}$. By [Yang et al., 2020, Lemma 17], we have $\max_{a \in \mathcal{A}} \langle a - A_{n,t}, \theta_n \rangle \le J\|\theta_n - \hat{\theta}_n\|^2/\|\theta_n\|$, where $A_{n,t} = \operatorname{argmax}_{a \in \mathcal{A}} \langle a, \hat{\theta}_n \rangle$. Thus,

$$
\begin{aligned}
R_\tau^n &= \mathbb{E}\left[ \tau \max_{a \in \mathcal{A}} \langle \theta_n, a \rangle - \sum_{t=1}^{\tau} \langle \theta_n, A_{n,t} \rangle \right] \\
&= \mathbb{E}\left[ \tau_2 \max_{a \in \mathcal{A}} \langle \theta_n, a \rangle - \sum_{t=1}^{\tau_2} \langle \theta_n, A_{n,t} \rangle + (\tau - \tau_2) \max_{a \in \mathcal{A}} \langle \theta_n, a \rangle - \sum_{t=\tau_2+1}^{\tau} \langle \theta_n, A_{n,t} \rangle \right] \\
&\le \lambda_0 \theta_{\max} \tau_2 + (\tau - \tau_2) J \frac{\mathbb{E}\|\theta_n - \hat{B}_n \hat{w}_n\|^2}{\theta_{\min}}.
\end{aligned}
$$

To bound the second term, we use Lemma 14 (subspace-informed estimation). We have

$$\mathbb{E}\left\| \hat{B}_n \hat{w}_n - \theta_n \right\|^2 \le \frac{m^2}{\lambda_0^2 \tau_2} + \|\hat{B}_{n,\perp}^\top \theta_n\|^2.$$

It follows that

$$R_\tau^n \le c \cdot \left( \tau_2 + \tau \cdot \left( \frac{m^2}{\tau_2} + \|\hat{B}_{n,\perp}^\top \theta_n\|_2^2 \right) \right).$$

In addition, when $\|\hat{B}_{n,\perp}^\top \theta_n\|_2 \le 2\alpha$,

$$R_\tau^n \le \lambda_0 \theta_{\max} \tau_2 + (\tau - \tau_2) \frac{J}{\theta_{\min}} \left( \frac{m^2}{\lambda_0^2 \tau_2} + 4\alpha^2 \right) \le 4c \left( \tau_2 + \tau \cdot \left( \frac{m^2}{\tau_2} + \alpha^2 \right) \right),$$

where $c = \max\left\{ \lambda_0 \theta_{\max}, \frac{J}{\lambda_0^2 \theta_{\min}}, \frac{J}{\theta_{\min}} \right\}$. $\qquad\square$

We now present Lemma 14 used in the proof above for subspace-informed estimation; see also [Qin et al., 2022, Lemma 2] and [Yang et al., 2020, Lemma 18].

**Lemma 14 (Subspace-informed estimation).** *Suppose Algorithm 2 is run on task $n$ with the exploration length $\tau_2$, then $\mathbb{E}\|\hat{\theta}_n - \theta_n\|^2 \le \frac{m^2}{\lambda_0^2 \tau_2} + \|\hat{B}_{n,\perp}^\top \theta_n\|^2$.*

*Proof.* Without loss of generality, we assume that $\tau_2$ is a multiple of $m$. Since $A_{n,t} = \lambda_0 \hat{B}_n(i)$, $i \in [m]$, and each action repeats $\lfloor \tau_2/m \rfloor$ times for $t \le \tau_2$, we have $\sum_{t=1}^{\tau_2} A_{n,t} A_{n,t}^\top = \frac{\tau_2 \lambda_0^2}{m} \hat{B}_n \hat{B}_n^\top$. Thus,

$$
\begin{aligned}
\sum_{t=1}^{\tau_2} \hat{B}_n^\top A_{n,t} A_{n,t}^\top \hat{B}_n &= \frac{\tau_2 \lambda_0^2}{m} \hat{B}_n^\top \hat{B}_n \hat{B}_n^\top \hat{B}_n \\
&= \frac{\tau_2 \lambda_0^2}{m} I_m.
\end{aligned}
$$

Then, the OLS estimator is given by

$$
\begin{aligned}
\hat{w}_n &= \left( \sum_{t=1}^{\tau_2} \hat{B}_n^\top A_{n,t} A_{n,t}^\top \hat{B}_n \right)^{-1} \sum_{t=1}^{\tau_2} \hat{B}_n^\top A_{n,t} r_{n,t} \\
&= \left( \sum_{t=1}^{\tau_2} \hat{B}_n^\top A_{n,t} A_{n,t}^\top \hat{B}_n \right)^{-1} \sum_{t=1}^{\tau_2} \hat{B}_n^\top A_{n,t} \left( A_{n,t}^\top B w_n + \eta_{n,t} \right) \\
&= \frac{m}{\tau_2 \lambda_0^2} \sum_{t=1}^{\tau_2} \hat{B}_n^\top A_{n,t} \left( A_{n,t}^\top B w_n + \eta_{n,t} \right) \\
&= \frac{m}{\tau_2 \lambda_0^2} \sum_{t=1}^{\tau_2} \hat{B}_n^\top A_{n,t} A_{n,t}^\top \left( \hat{B}_n \hat{B}_n^\top + \hat{B}_{n,\perp} \hat{B}_{n,\perp}^\top \right) B w_n + \frac{m}{\tau_2 \lambda_0^2} \sum_{t=1}^{\tau_2} \hat{B}_n^\top A_{n,t} \eta_{n,t} \\
&\hspace{6cm} (\hat{B}_n \hat{B}_n^\top + \hat{B}_{n,\perp} \hat{B}_{n,\perp}^\top = I) \\
&= \hat{B}_n^\top B w_n + \frac{m}{\tau_2 \lambda_0^2} \sum_{t=1}^{\tau_2} \hat{B}_n^\top A_{n,t} A_{n,t}^\top \hat{B}_{n,\perp} \hat{B}_{n,\perp}^\top B w_n + \frac{m}{\tau_2 \lambda_0^2} \sum_{t=1}^{\tau_2} \hat{B}_n^\top A_{n,t} \eta_{n,t} \\
&= \hat{B}_n^\top B w_n + \frac{m}{\tau_2 \lambda_0^2} \sum_{t=1}^{\tau_2} \hat{B}_n^\top A_{n,t} \eta_{n,t}, \hspace{3cm} (A_{n,t}^\top \hat{B}_{n,\perp} = 0)
\end{aligned}
$$

where the first equality uses the closed-form solution of OLS; the second equality is by the definition of $r_{n,t}$; and the other equalities follow from algebraic manipulations.

Now, we have

$$
\begin{aligned}
\hat{\theta}_n - \theta_n &= \hat{B}_n \hat{w}_n - B w_n \\
&= \hat{B}_n \left( \hat{B}_n^\top B w_n + \frac{m}{\tau_2 \lambda_0^2} \sum_{t=1}^{\tau_2} \hat{B}_n^\top A_{n,t} \eta_{n,t} \right) - B w_n \\
&= \underbrace{\left( \hat{B} \hat{B}_n^\top B w_n - B w \right)}_{=:s_1} + \underbrace{\frac{m}{\tau_2 \lambda_0^2} \sum_{t=1}^{\tau_2} \hat{B}_n \hat{B}_n^\top A_{n,t} \eta_{n,t}}_{=:s_2} .
\end{aligned}
$$

For $s_1$, we have

$$
\begin{aligned}
\|s_1\|^2 &= \|\hat{B}_n \hat{B}_n^\top B w_n - B w_n\|^2 \\
&= \|(I - \hat{B}_{n,\perp} \hat{B}_{n,\perp}^\top) B w_n - B w_n\|^2 \\
&= \|\hat{B}_{n,\perp} \hat{B}_{n,\perp}^\top B w_n\|^2 \\
&\leq \|\hat{B}_{n,\perp}\|_{op}^2 \|\hat{B}_{n,\perp}^\top \theta_n\|^2 \\
&\leq \|\hat{B}_{n,\perp}^\top \theta_n\|^2
\end{aligned}
$$

For $s_2$, we have

$$\mathbb{E}\|s_2\|^2 = \mathbb{E}\left\|\frac{m}{\tau_2\lambda_0^2}\sum_{t=1}^{\tau_2}\hat{B}_n\hat{B}_n^\top A_{n,t}\eta_{n,t}\right\|^2$$

$$= \frac{m^2}{\tau_2^2\lambda_0^4}\mathbb{E}\left[\sum_{t=1}^{\tau_2}\left(\hat{B}_n\hat{B}_n^\top A_{n,t}\eta_{n,t}\right)^\top \hat{B}_n\hat{B}_n^\top A_{n,t}\eta_{n,t}\right]$$

$$= \frac{m^2}{\tau_2^2\lambda_0^4}\mathbb{E}\|\eta_{n,t}\|^2\sum_{t=1}^{\tau_2}A_{n,t}^\top\hat{B}_n\hat{B}_n^\top A_{n,t}$$

$$= \frac{m^2}{\tau_2^2\lambda_0^4}\mathbb{E}\|\eta_{n,t}\|^2\sum_{t=1}^{\tau_2}A_{n,t}^\top\left(I_d - \hat{B}_{n,\perp}\hat{B}_{n,\perp}^\top\right)A_{n,t} \qquad (\hat{B}_n\hat{B}_n^\top + \hat{B}_{n,\perp}\hat{B}_{n,\perp}^\top = I)$$

$$= \frac{m^2}{\tau_2^2\lambda_0^4}\mathbb{E}\|\eta_{n,t}\|^2\sum_{t=1}^{\tau_2}A_{n,t}^\top A_{n,t} \qquad (A_{n,t}^\top\hat{B}_{n,\perp} = 0)$$

$$= \frac{m^2}{\tau_2^2\lambda_0^4}\mathbb{E}\|\eta_{n,t}\|^2\tau_2\lambda_0^2$$

$$= \frac{m^2}{\tau_2\lambda_0^2}\mathbb{E}\|\eta_{n,t}\|^2$$

$$\leq \frac{m^2}{\tau_2\lambda_0^2}. \qquad (\eta_{n,t} \text{ is 1-sub-Gaussian noise assumption})$$

Hence,

$$\mathbb{E}\|\hat{\theta}_n - \theta_n\|^2 \leq \mathbb{E}\|s_1\|^2 + \mathbb{E}\|s_2\|^2 \qquad \text{(Triangle inequality)}$$

$$\leq \frac{m^2}{\tau_2\lambda_0^2} + \|\hat{B}_{n,\perp}^\top\theta_n\|^2. \qquad \square$$

## F   Proof of lemma 6: Regret of the subspace selection game

**Lemma 6.** *Let $\varepsilon = \alpha = c_2 d\sqrt{\frac{\ln\frac{d}{\delta}}{\tau_1}}$ (with $c_2$ defined in Lemma 3) and $\delta = \frac{1}{N^2}$, where $c$ is a constant in Lemma 13. Then, assuming that $\tau \gg d^2$, Algorithm 3 chooses a sequence of subspaces $\left\{\hat{B}_n\right\}$ over the expert set $\mathcal{E}^\varepsilon$, defined in Definition 5, such that:*

$$\sum_{n=1}^N \mathbb{E}\left[C_n(\hat{B}_n)\right] \leq O\left(N\boldsymbol{C_{hit}} + \frac{\boldsymbol{C_{miss}}\log|\mathcal{E}^\varepsilon|}{p}\right) = \tilde{O}\left(N\left(\tau_2 + \tau\cdot\left(\frac{m^2}{\tau_2} + \alpha^2\right)\right) + \frac{\tau dm}{p}\right).$$

*Proof.* Recall the guarantee of EWA from Freund and Schapire [1997]:

**Theorem 15** (Freund and Schapire [1997])**.** *For any sequence of loss vectors $\ell_1, \cdots, \ell_T$ and the initial weights of the experts are $w_1^i = 1/n$ for all $i \in [n]$ and $\gamma \in (0,1)$, the EWA algorithm with learning rate $\eta = -\ln(1-\gamma)$ generates $\{p_t\}_{t=1}^T$ such that its expected loss is bounded by*

$$\sum_{t=1}^T \langle \ell_t, p_t\rangle \leq \frac{-\log(1-\gamma)}{\gamma}\ell_t(i) + \frac{\ln n}{\gamma}, \quad \forall i = 1,\ldots,n;$$

*specifically, if at each round we choose $i_t \sim p_t$,*

$$\mathbb{E}\left[\sum_{t=1}^T\ell_t(i_t)\right] \leq -\frac{\ln(1-\gamma)}{\gamma}\cdot\mathbb{E}\left[\sum_{t=1}^T\ell_t(i)\right] + \frac{\ln n}{\gamma}, \quad \forall i = 1,\ldots,n.$$

Applying Theorem 15 with $T = N$, expert set $\mathcal{E}^\varepsilon$, $i_t = \hat{B}_n$, loss functions $\ell_n(B) = \frac{p}{\mathbf{C}_{\mathtt{miss}}}\left[\tilde{C}_n(B)\frac{Z_n}{p} - \mathbf{C}_{\mathtt{hit}}\frac{Z_n}{p}\right]$, $n \in [N]$, $B \in \mathcal{E}^\epsilon$, and the baseline expert $i = B_\varepsilon$, we get:

$$\mathbb{E}\left[\sum_{n=1}^{N}\ell_n(\hat{B}_n)\right] \leq -\frac{\ln(1-\gamma)}{\gamma}\cdot\mathbb{E}\left[\sum_{n=1}^{N}\ell_t(B_\varepsilon)\right] + \frac{\ln|\mathcal{E}^\varepsilon|}{\gamma}.$$

Using the basic fact that $\frac{-\ln(1-x)}{x} \leq 1 + x$ for $x \in (0, \frac{1}{2}]$ we have that, for $\gamma \in (0, \frac{1}{2}]$,

$$\mathbb{E}\left[\sum_{n=1}^{N}\ell_n(\hat{B}_n)\right] \leq (1+\gamma)\mathbb{E}\left[\sum_{n=1}^{N}\ell_n(B_\varepsilon)\right] + \frac{\log|\mathcal{E}^\varepsilon|}{\gamma}.$$

Define $F_n = \left\{Z_n = 0 \vee (Z_n = 1 \wedge \|\hat{\theta}_n - \theta_n\| \leq \alpha)\right\}$ and $F = \cap_{n=1}^{N}F_n$. Then, when all of our estimations $\hat{\theta}_n$ are accurate, we have: $\mathbb{I}(F)\sum_{n=1}^{N}\ell_n(B_\varepsilon) = 0$ for $\ell_n(B) = \frac{p}{\mathbf{C}_{\mathtt{miss}}}\left[\bar{C}_n(B) - \mathbf{C}_{\mathtt{hit}}\frac{Z_n}{p}\right]$ and $B_\varepsilon \in \mathcal{E}^\varepsilon$.

By the definition of $F_n$, we have: $F_n^c = \left\{Z_n = 1 \wedge \|\hat{\theta}_n - \theta_n\| > \alpha\right\}$. Thus, by Lemma 3, $P(F_n^c) = P(\|\hat{\theta}_n - \theta_n\| > \alpha \mid Z_n = 1)P(Z_n = 1) \leq p\delta$, and therefore $P(F) \geq 1 - \sum_{n=1}^{N}P(F_n^c) = 1 - Np\delta$. Since Algorithm 3 chooses the learning rate for EWA as $\eta = \log(2)$, thus, $\gamma = 1 - \exp(-\eta) = 0.5$, we have

$$\begin{aligned}
\mathbb{E}\left[\sum_{n=1}^{N}\ell_n(\hat{B}_n) - \sum_{n=1}^{N}\ell_n(B_\varepsilon)\right] &\leq \mathbb{E}\left[\gamma\sum_{n=1}^{N}\ell_n(B_\varepsilon) + \frac{\log|\mathcal{E}^\varepsilon|}{\gamma}\right] \\
&= \mathbb{E}\left[\mathbb{I}(F)\gamma\sum_{n=1}^{N}\ell_n(B_\varepsilon) + \mathbb{I}(F^c)\gamma\sum_{n=1}^{N}\ell_n(B_\varepsilon) + \frac{\log|\mathcal{E}^\varepsilon|}{\gamma}\right] \\
&= \mathbb{E}\left[0 + \mathbb{I}(F^c)\gamma\sum_{n=1}^{N}\ell_n(B_\varepsilon) + \frac{\log|\mathcal{E}^\varepsilon|}{\gamma}\right] \\
&\leq \mathbb{E}\left[\mathbb{I}(F^c)\gamma N + \frac{\log|\mathcal{E}^\varepsilon|}{\gamma}\right] \\
&\leq N^2\gamma\delta + \frac{\log|\mathcal{E}^\varepsilon|}{\gamma} \qquad\qquad (p \leq 1) \\
&\leq O\left(\log|\mathcal{E}^\varepsilon|\right). \qquad\qquad (\delta \leq \frac{4\log|\mathcal{E}^\varepsilon|}{N^2} \text{ and } \gamma = 1/2)
\end{aligned}$$

Then,

$$\mathbb{E}\left[\sum_{n=1}^{N}\ell_n(\hat{B}_n) - \sum_{n=1}^{N}\ell_n(B_\varepsilon)\right] \leq O(\log|\mathcal{E}^\varepsilon|)$$

$$\implies \mathbb{E}\left[\sum_{n=1}^{N}\ell_n(\hat{B}_n)\right] - \mathbb{E}\left[\mathbb{I}(F)\sum_{n=1}^{N}\ell_n(B_\varepsilon) + \mathbb{I}(F^c)\sum_{n=1}^{N}\ell_n(B_\varepsilon)\right] \leq O(\log|\mathcal{E}^\varepsilon|)$$

$$\implies \mathbb{E}\left[\sum_{n=1}^{N}\ell_n(\hat{B}_n)\right] - \mathbb{E}\left[\mathbb{I}(F^c)\sum_{n=1}^{N}\ell_n(B_\varepsilon)\right] \leq O(\log|\mathcal{E}^\varepsilon|)$$

$$\implies \mathbb{E}\left[\sum_{n=1}^{N}\ell_n(\hat{B}_n)\right] \leq O(\log|\mathcal{E}^\varepsilon|) + N^2\delta$$

$$(p \leq 1)$$

$$\implies \sum_{n=1}^{N}\frac{p}{\mathbf{C}_{\texttt{miss}}}\mathbb{E}\left[\bar{C}_n(\hat{B}_n) - \mathbf{C}_{\texttt{hit}}\frac{Z_n}{p}\right] \leq O(\log|\mathcal{E}^\varepsilon|)$$

$$(N^2\delta \leq O(\log|\mathcal{E}^\varepsilon|))$$

$$\implies \sum_{n=1}^{N}\mathbb{E}\left[\bar{C}_n(\hat{B}_n)\right] - N\mathbf{C}_{\texttt{hit}} \leq O\left(\frac{\mathbf{C}_{\texttt{miss}}\log|\mathcal{E}^\varepsilon|}{p}\right)$$

$$\implies \sum_{n=1}^{N}\mathbb{E}\left[\bar{C}_n(\hat{B}_n)\right] \leq O\left(N\mathbf{C}_{\texttt{hit}} + \frac{\mathbf{C}_{\texttt{miss}}\log|\mathcal{E}^\varepsilon|}{p}\right).$$

Thus,

$$\mathbb{E}\left[\bar{C}_n(\hat{B}_n)\right] = \mathbb{E}\left[\tilde{C}_n(\hat{B}_n)\cdot\frac{Z_n}{p}\right]$$

$$\geq \mathbb{E}\left[\tilde{C}_n(\hat{B}_n)\cdot\mathbb{I}(F_n)\cdot\frac{Z_n}{p}\right]$$

$$\geq \mathbb{E}\left[C_n(\hat{B}_n)\cdot(1 - \mathbb{I}(F_n^c))\cdot\frac{Z_n}{p}\right] \quad (\tilde{C}_n(B)\mathbb{I}(F_n) \geq C_n(B)\mathbb{I}(F_n) \text{ in Lemma 11})$$

$$= C_n(\hat{B}_n) - \mathbb{E}\left[C_n(\hat{B}_n)\cdot\mathbb{I}(F_n^c)\cdot\frac{Z_n}{p}\right]$$

$$\geq C_n(\hat{B}_n) - \frac{\mathbf{C}_{\texttt{miss}}}{N^2},$$

where, for the last inequality, we use the observation that $\mathbb{E}\left[C_n(\hat{B}_n)\cdot\mathbb{I}(F_n^c)\cdot\frac{Z_n}{p}\right] \leq \frac{\mathbf{C}_{\texttt{miss}}}{p}\mathbb{E}\left[\mathbb{I}(F_n^c)\right] \leq \frac{\mathbf{C}_{\texttt{miss}}}{p}P(F_n^c) \leq \mathbf{C}_{\texttt{miss}}\delta \leq \frac{\mathbf{C}_{\texttt{miss}}}{N^2}$, since $P(F_n^c) \leq p\delta \leq \frac{p}{N^2}$.

Hence,

$$\sum_{n=1}^{N}\mathbb{E}\left[\bar{C}_n(\hat{B}_n)\right] \leq O\left(N\mathbf{C}_{\texttt{hit}} + \frac{\mathbf{C}_{\texttt{miss}}\log|\mathcal{E}^\varepsilon|}{p}\right)$$

$$\implies \sum_{n=1}^{N}\mathbb{E}\left[C_n(\hat{B}_n)\right] \leq O\left(N\mathbf{C}_{\texttt{hit}} + \frac{\mathbf{C}_{\texttt{miss}}\log|\mathcal{E}^\varepsilon|}{p} + N\cdot\frac{\mathbf{C}_{\texttt{miss}}}{N^2}\right)$$

$$= O\left(N\mathbf{C}_{\texttt{hit}} + \frac{\mathbf{C}_{\texttt{miss}}\log|\mathcal{E}^\varepsilon|}{p}\right). \quad (1 \leq O(\log|\mathcal{E}^\varepsilon|) \text{ and } \frac{1}{N} \leq 1 \leq \frac{1}{p})$$

Substituting $\mathbf{C}_{\texttt{hit}}$ and $\mathbf{C}_{\texttt{miss}}$ in Equation (2) to complete the proof. $\qquad\square$

# G Theorem 7: meta-regret guarantee

**Theorem 7.** *With exploration probability* $p = \min\left(\left(\frac{2m\sqrt{\tau}}{N}\right)^{\frac{2}{3}}, 1\right)$*, by choosing* $\varepsilon = \alpha = c_2 d\sqrt{\frac{\ln\frac{d}{\delta}}{\tau_1}}$ *(with* $c_2$ *defined in Lemma 3) , where* $\delta = \frac{1}{N^2}$*,* $\tau_1 = d \cdot \left\lfloor \min\left(d\sqrt{\frac{\tau}{p}}, \tau\right)/d \right\rfloor$*,* $\tau_2 = m \cdot \lfloor\sqrt{\tau}\rfloor$*, the meta-regret of the* BOSS *algorithm is bounded by:*

$$R_\tau \leq \tilde{O}\left(Nm\sqrt{\tau} + N^{\frac{2}{3}}\tau^{\frac{2}{3}}dm^{\frac{1}{3}} + Nd^2 + \tau md\right). \tag{5}$$

*Remainder of the Proof of Theorem 7.* Recall that in the proof sketch of Theorem 7 (Section 4), we have proved that

$$R_\tau \leq \tilde{O}\left(Np\tau_1 + N\tau\frac{d^2}{\tau_1} + \frac{\tau dm}{p} + N\tau_2 + N\tau\frac{m^2}{\tau_2}\right).$$

Now, by the choice of $\tau_2 = m \cdot \lfloor\sqrt{\tau}\rfloor$,

$$R_\tau \leq \tilde{O}\left(Nm\sqrt{\tau} + N\tau\frac{d^2}{\tau_1} + Np\tau_1 + \frac{\tau dm}{p}\right).$$

We want to tune the parameters $p, \tau_1$ to minimize the meta-regret subjected to the constraint: $p \in [0, 1]$ and $\tau_1 \in [0, \tau]$.

The meta-regret is:

$$
\begin{aligned}
R_\tau &\leq \tilde{O}\left(Nm\sqrt{\tau} + N\tau\frac{d^2}{\tau_1} + Np\tau_1 + \frac{\tau dm}{p}\right) \\
&= \tilde{O}\left(Nm\sqrt{\tau} + Nd\sqrt{\tau p} + \frac{\tau dm}{p} + Nd^2\right) \qquad \text{(Choose } \tau_1 = d \cdot \left\lfloor \min\left(d\sqrt{\tfrac{\tau}{p}}, \tau\right)/d \right\rfloor\text{)} \\
&= \tilde{O}\left(Nm\sqrt{\tau} + N^{\frac{2}{3}}\tau^{\frac{2}{3}}dm^{\frac{1}{3}} + Nd^2 + \tau md\right). \qquad \text{(Choose } p = \min\left(\left(\tfrac{2m\sqrt{\tau}}{N}\right)^{\frac{2}{3}}, 1\right)\text{)}
\end{aligned}
$$

$\square$

# H Additional experiment result

## H.1 Adversarial environment for SeqRepL's deterministic exploration schedule

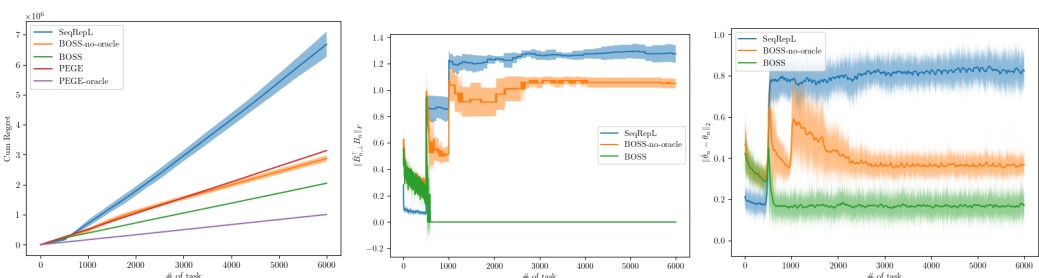

Figure 2: Comparing the cumulative regret of BOSS and other baselines. The setting is $(N, \tau, d, m) = (6000, 2000, 10, 3)$ and $\|\theta_n\|_2 \in [0.8, 1]$ $\forall n \in [N]$ chosen uniformly at random from this interval. SeqRepL, BOSS, and BOSS-no-oracle uses the same hyperparameters $\tau_1 = 400, \tau_2 = 50$. The environment only reveals a new subspace dimension at tasks 1, 501, and 1001, and only reveals the same dimension at Qin et al. [2022]'s deterministic exploration schedule.

## H.2 When the Task Diversity assumption is satisfied

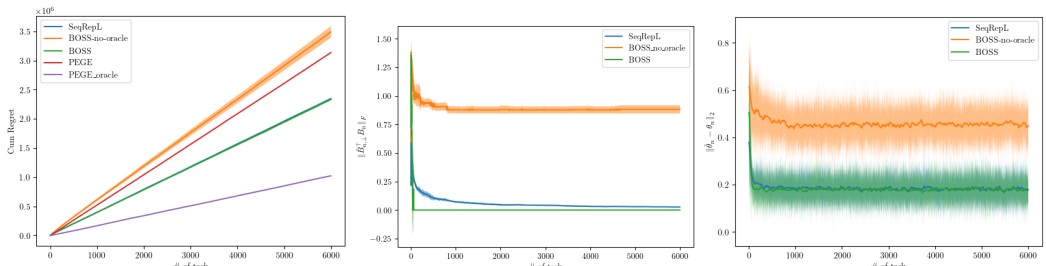

Figure 3: Comparing the cumulative regret of BOSS and other baselines. The setting is $(N, \tau, d, m) = (6000, 2000, 10, 3)$ and $\|\theta_n\|_2 \in [0.8, 1] \; \forall n \in [N]$. SeqRepL, BOSS, and BOSS-no-oracle uses the same hyperparameters $\tau_1 = 1000, \tau_2 = 300$. The task diversity assumption is satisfied: each $\theta_n$ is generated by a linear combinations of the columns in $B_n$ – the subspace spanning $\theta_1, \cdots, \theta_{n-1}$. The performance of SeqRepL and BOSS is almost identical in the left figure.

