# OpenReview forum: "Beyond task diversity: provable representation transfer for sequential multitask linear bandits"
_NeurIPS.cc/2024/Conference — NeurIPS 2024 poster_

### Official Review · Reviewer_u72q · 2024-06-23

**Soundness:** 3
**Presentation:** 3
**Contribution:** 2
**Rating:** 5
**Confidence:** 3

**Summary:**

This paper extends the existing work on multi-task linear bandits where the task parameters lie in a low rank subspace. Specifically, this paper assumes $N$ $d$-dimensional linear bandits with parameters lie in an $m$-dimensional subspace. For such a setting, the classical approaches yield a regret linear in $N$ and $d$. This paper then proposed a new algorithm that improves the regret with mild assumptions of well-conditioned ellipsoids of the tasks and bounded task parameters. The algorithm is based on a reduction to a bandit online subspace selection problem. The key idea of the proposed approach is bi-level. At the lower level, the algorithm performs either an meta-exploration algorithm or a meta-exploitation algorithm for each of the tasks. At the upper level, the learner either chooses exploration/exploitation or chooses a subspace to use for exploitation. The effectiveness of the proposed algorithm is verified empirically on synthetic adversarial settings.

**Strengths:**

**Significance**: This paper proposed a new algorithm for multi-task linear bandit with provable low regret without making strong assumptions on task parameters. The sequential setting studied is more challenging than parallel settings due to the slow revelation of the underlining subspace. Compared with a few related existing works, this paper does not need assumptions such as task diversity.

**Quality**: this paper is in general good quality. The algorithm is well described.
The theoretical part of the work is clearly formulated. Thought the experiments are synthetic, the result still verifies the effectiveness of the proposed algorithm.

**Clearity**: The design of the algorithm and intuition behind the design is clearly explained.

**Weaknesses:**

There is no major weakness in the paper. I do have some questions regarding the following:

**1**. This paper assumes knowledge of the parameters corresponding to the task number, the dimension, and the number of rounds, which is not ideal.

**2**. [Originality] The algorithmic design and the guarantee is closely related to a a few works, including the PEGE by [Rusmevichientong and Tsitsiklis 2010] (related to lemma 3 in this paper), the analysis in Yang et al [2020] (related to lemma 4), EWA algorithm (related to lemma 5). Therefore, the technical novelty of the proposed algorithm and analysis is not very obvious to me.

**Questions:**

Please see weaknesses.


Also, I am curious about the following:

1. This paper assumes knowledge to a few hyper-parameters. Among these, I wonder whether the assumption on the prior knowledge of the $\tau$ parameter can be relaxed with certain adaptive design. Could the author elaborate on why knowledge is necessary for the current algorithm to work?

2. Is there a corollary or simple extension that can readily extend the current result to the parallel setting?

**Limitations:**

Yes.

---

> ### Author Rebuttal · Authors · 2024-08-07
>
> - Originality: The algorithmic design and the guarantee is closely related to a a few works, including the PEGE by [Rusmevichientong and Tsitsiklis 2010] (related to lemma 3 in this paper), the analysis in Yang et al [2020] (related to lemma 4), EWA algorithm (related to lemma 5). Therefore, the technical novelty of the proposed algorithm and analysis is not very obvious to me.
>
> $=>$ As discussed in the Introduction and Table 1, Table 2, our work fills the gap of the sequential representation transfer for linear bandit, without the Task Diversity assumption. Before our work, no algorithms could provably beat the naive individual single-task baseline. The absence of task diversity assumption makes meta-learning the task representation matrix $B$ nontrivial.
>
> Our proposal designs a bi-level approach for this problem and shows that, under mild assumptions, we can efficiently learn the underlying subspace in an online sense and maintain the correct uncertainty over possible experts when the environment acts adversarially (i.e. not revealing the full underlying subspace till the end). We believe our reduction to online subspace selection is new to sequential representation transfer and may benefit other meta-learning applications such as in RL or supervised learning.

---

> > ### Comment · Reviewer_u72q · 2024-08-12
> > **Response to rebuttal**
> >
> > I thank you authors for their efforts in the rebuttal. I will take all the rebuttals into consideration during the discussion period between reviewers.

---

### Official Review · Reviewer_vp1F · 2024-07-12

**Soundness:** 2
**Presentation:** 3
**Contribution:** 3
**Rating:** 5
**Confidence:** 3

**Summary:**

The paper studies lifelong learning in linear bandits and designs a two-level algorithm with provable low regret without task diversity assumptions. The paper assumes the tasks share a low-rank representation and provides a regret upper bound.

**Strengths:**

The paper uses a two-level approach to solve this problem, which is novel. A theoretical guarantee is given with numerical experiments performed.

**Weaknesses:**

There are some statements in the theorems that seem wrong and the upper bound provided does not seems tight. See Questions for details.

**Questions:**

1) In line 109, for a matrix $U$, isn't $U_{\perp}$ not unique? This should be defined as a set or stated as one of the matrices.
2) The Assumption 2 is stated as linear bandits. I didn't get the reason for this title.
3) In line 3 in Algorithm 1, it should be $\mu=\frac{\tau_1}{d}$
4) The statement in Lemma 4 seems problematic. As stated in 1), the matrix $\hat B_{n,\perp}$ is not unique, hence the RHS is a matrix-dependent bound. Thus, the condition $||\hat B_{n,\perp} \theta_{n}||_2\leq 2\alpha$ is also dependent on the choice of the matrix. This discussion is missing in the paper. This will also affect the definitions in (2) and (3), and the following results.
5) In line 173, the definition of the expert set $\mathcal{E}^{\epsilon}$ is in the Appendix, which makes the paper not self-contained.
6) What does the EWA algorithm do in Algorithm 3? Which part of the results is directly from the EWA algorithm?
7) Overall, the paper provides the regret bounds of order  $\tilde{\mathcal{O}} (N m\sqrt{\tau} + N^{\frac{3}{2}} \tau^{\frac{2}{3}} d m^{\frac{1}{3}} + \tau m d)$ in equation (5),  times of interactions. Compared to the baseline, which is $\tilde{\mathcal{O}} (N d \sqrt{\tau})$, I didn't see a clear improvement in the regret/ Firstly, the regret scales linear with $\tau$ when fix $N$, which is known to be suboptimal in linear bandit. At least some truncation can be done in the first $md$ tasks to get rid of the linear term. This term cannot be hidden from the abstract from my point of view. Secondly, to achieve similar upper bounds, one needs $N\geq md$ and $d^2\leq\tau\leq (\frac{N}{m})^2$, which means the algorithm can only learn for a finite time and the tightness discussion is missing the paragraph Comparison with lower bounds with only a conjecture.
7) The author answer Yes to the Question about Open access to data and code in the Checklist, but the code is not released during the submission. And I am curious how Figure 1B is generated.
9) The labels on the y-axis in the figures need to be fixed.

**Limitations:**

The regret upper bounds has a hugh gap to the lower bound.

---

> ### Author Rebuttal · Authors · 2024-08-07
>
> - ``In line 109, for a matrix $U$ , isn’t $U_{\perp}$ not unique? This should be defined as a set or stated as one of the matrices.''
>
> **We thank the reviewer for pointing out this impreciseness.** Indeed, the choice of $U_\perp$ is not unique according to our writing in the submitted version.
>
> To make the definition well-defined, it suffices to fix any orthonormal basis $U_\perp$; one may break ties in dictionary order. We will clarify this in the final version of the paper.
>
> Indeed, the specific choice of $U_\perp$ does not affect the correctness of our proof -- e.g., $ \| \hat{B}_{n, \perp} \theta_n \|_2$ are always the same regardless
>
> of the specific choice of $\hat{B}_{n, \perp}$, as long as its columns form a orthonormal basis of $\text{span}(\hat{B}_n)^\perp$:
>
> For any two choices of $\hat{B}_{n, \perp}$
>
> (denoted by $\hat{B}_{\perp, 1}$,
>
> $\hat{B}_{\perp, 2}$, respectively)
>
> there exists orthonormal $V \in \mathbb{R}^{{(d-m) \times (d-m)}}$ such that
>
> $\hat{B}_{\perp, 1}$
>
> $ = \hat{B}_{\perp, 2} V$.
>
> Therefore,
>
> $\|\hat{B}_{\perp, 1}^{\top} \theta_n\|_2$
>
> $= \|V^{\top}\hat{B}_{\perp, 2}^{\top} \theta_n\|_2$
>
> $ = \|\hat{B}_{\perp, 2}^{\top} \theta_n\|_2$
>
> To show that all valid choices of $\hat{B}_{n, \perp}$  are equivalent up to a $(d-m) \times (d-m)$ orthogonal transformation, we have:
>
> Let $W$ be a $k$-dimensional subspace of $\mathbb{R}^d$. Let $B, \hat{B} \in \mathbb{R}^{d \times k}$ be matrices whose columns form an orthonormal basis of $W$. Then, there exists an orthogonal matrix $V \in \mathbb{R}^{k \times k}$ such that $\hat{B} = B V$.
>
> **Proof:** Since $B$ is a basis of $W$, there exists some $V$ such that $\hat{B} = BV$. Since $V^\top V = V^\top (B^\top B) V = (BV)^\top (BV) = \hat{B}^\top \hat{B} = I$, $V$ is an orthogonal matrix.
>
>
>
> - ``The statement in Lemma 4 seems problematic. As stated in 1), the matrix $\hat{B}_{n,\perp}$ is not unique, hence the RHS is a matrix-dependent bound.
>
> Thus, the condition $||\hat{B}_{n,\perp} \theta_n||_2 \leq 2 \alpha$ is also dependent on the choice of the matrix. This discussion is missing in the paper. This will also affect the definitions in (2) and (3), and the following results.''
>
>
> $=>$ As we mention in our previous item, the value of $|| \hat{B}_{n, \perp}^\top \theta_n ||$ is invariant to the specific choice of matrix
>
> $\hat{B}_{n, \perp}$. Still, we agree that this can be made clearer in the main paper, so we will make the necessary edits in the final version.
>
> - ``What does the EWA algorithm do in Algorithm 3? Which part of the results is directly from the EWA algorithm?''
>
> $=>$ The algorithm uses Exponentially Weighted Algorithm (EWA) in line 8 of Algorithm 3. The intuition of using EWA is to choose subspaces $\hat{B}_n$ online such that their linear spans capture $\theta_n$'s over tasks $n=1,..., N$. The idea is to use the feedback from the meta-exploration tasks to update the weights of all possible experts from the expert set. Since this expert set $\alpha$-cover the true $B$, the EWA guarantee would ensure that BOSS would efficiently learn the closest expert to the true $B$ while maintaining the correct uncertainty over all possible experts when the subspace dimensions are not fully revealed, thus, neatly dealing with the non-Task Diversity setting.
>
> Specifically speaking, the EWA's guarantee is used in the proof of Lemma 5, line 455. We will provide a full description of the algorithm with weight updates in the Appendix in the final version.
>
> - ``Overall, the paper provides the regret bounds of order  $\tilde{O} (N m\sqrt{\tau} + N^{3/2} \tau^{2/3} dm^{1/3} + \tau md)$ in equation (5), times of interactions. Compared to the baseline, which is  $\tilde{O}(N d\sqrt{\tau} )$, I didn’t see a clear improvement in the regret. ... the regret scales linear with when fix N, which is known to be suboptimal in linear bandit. At least some truncation can be done in the first tasks to get rid of the linear term. This term cannot be hidden from the abstract from my point of view''
>
> **We agree that the $\tau m d$ term cannot be ignored and will add it to the abstract.** Note that we are studying the multi-task problem; thus, the interesting regime is when the number of tasks $N$ is large enough to allow useful transfer learning across tasks. The $\tau md$ part of the regret can be understood as the learner ``sacrifices'' a small number of tasks to learn transferable knowledge. This last term is of lower order than $N^{2/3} \tau^{2/3} d m^{1/3}$ when $\tau \ll (\frac{N}{m})^2$.
>
> - ``The definition of the expert set $\mathcal{E}^{\varepsilon}$ is in the Appendix, which makes the paper not self-contained''
>
> **We agree.** At first, we moved these definitions to the appendix to not distract the reader. We realized that they may be required for the main paper to be self-contained, so we will make the necessary edits.
>
> - ``... the code is not released during the submission. And I am curious how Figure 1B is generated.''
>
> **Correct**. We want to add some additional experiment results. The code is in https://anonymous.4open.science/r/Serena-C5F1.
>
> For Fig 1b, since we use a simulated environment, we have access to the $B_n$, the subspace spanned by  $\left ( \theta_i \right )_{i=1}^n$ thus far and the estimated subspace $\hat{B}_n$ of the learners for each task. Thus, we plot this figure on this information.
>
> - ``The Assumption 2 is stated as linear bandits. I didn’t get the reason for this title.''
>
> $=>$ We will change it to ``linear bandits with ellipsoid action sets'' in the final version.

---

> > ### Comment · Reviewer_vp1F · 2024-08-12
> >
> > Thanks for clarifying some of the points I over-interpreted. I would like to raise the score to 5 but reduce the confidence to 3. The main weakness in
> > my opinion is still the linear term in $\tau$. Consider any real-world tasks (such as search systems or recommender systems like movie recommendations), there will be a large volume of queries and $\tau$ will be arbitrarily larger than poly$(N)$.

---

### Official Review · Reviewer_9RaC · 2024-07-12

**Soundness:** 3
**Presentation:** 2
**Contribution:** 3
**Rating:** 5
**Confidence:** 4

**Summary:**

The paper proposes a strategy for the multi task linear bandit problem, where the tasks are assumed to share a low rank representation, which is ought to be learnt via a new meta learning strategy, where meta exploration and exploitation needs to be balanced. The low rank representation is learnt via optimizing a newly constructed loss function over an epsilon-net. The tasks appear sequentially to the learner and the authors explicitly do not make any assumption on the task diversity of their setting.
The authors provide a meta regret bound for their algorithms and synthetic data results.

**Strengths:**

The paper is well written and easy to follow

Original idea for solving the multi-task/meta learning setting with low rank representation

The assumption on the action set is weaker than in comparable works (constant action set vs iid action set)

**Weaknesses:**

The EWA algorithm for the meta learning procedure should not just be referenced but actually appear in the paper (or at least in the supplementary)

The presentation of the actual meta learning could be made clearer. For that matter, definition 7 and 8 should be moved to the main paper so it becomes clearer what the uniform distributions D_n are supposed to be.

The experiments would benefit if there were another baseline to compare to (Cella et al or Bilaj et al as they appear in the related work section)

The plot labels of figure 1 b) and c) should have latex style writing

**Questions:**

In Figure 1a) it looks like the Boss algorithm performs worse after a new dimension for the subspace is incremented at n= 2501, does that mean that the dimension of the subspace was falsely estimated?

at n=1 the the low rank should be estimated as m=1 as well, which makes it hard to minimize $B^T_{n+1,\perp}\theta_{n+1}$, (Since the n+1th task parameter might be outside the subspace selected for the first n tasks), Intuitively, the algorithm should work best if at least m task parameters are already bias-less estimated for a m-dimensional low rank structure. Could you explain how you mitigate this issue?

Shouldn't a bound on the term $||B_{\perp}-\hat{B}_{n+1,\perp}||$ or a term that showcases the estimation error on the representation,  appear in the final regret?

Does p need to be constant? after enough tasks explored, p could gradually vanish. I see in the final bound that $p \sim 1/\sqrt[3]{N^2}$, which is still constant.

**Limitations:**

Aside from the assumptions on their setting made, I did not see any limitations explicitly addressed.

---

> ### Author Rebuttal · Authors · 2024-08-07
>
> - ``Shouldn’t a bound on the term $||B_{\perp} -\hat{B}_{n+1,\perp}||$ or a term that showcases the estimation error on the representation, appear in the final regret?''
>
> **Yes.** Indeed, our regret bound has an implicit dependence on $|| \hat{B}_{n, \perp}^T \theta_n ||$ - see Eq. (2).
>
> This is a key quantity like the $||B_{\perp} -\hat{B}_{n+1,\perp}||$ you mentioned.
>
> - ``At n=1 the the low rank should be estimated as m=1 as well, which makes it hard to minimize $||B^T_{n+1,\perp} \theta_{n+1}||$ , (Since the n+1th task parameter might be outside the subspace selected for the first n tasks), Intuitively, the algorithm should work best if at least m task parameters are already bias-less estimated for a m-dimensional low rank structure. Could you explain how you mitigate this issue?''
>
> $=>$ If we understand correctly, the reviewer was referring to that the algorithm needs to overcome additional challenges without the task diversity assumption. Without task diversity, we cannot hope to obtain estimate $\hat{B}_n$'s that converge to the underlying $B$. Instead, we use online learning with randomized meta-exploration (using the learned $\hat{\theta}_n$'s to estimate the underlying representation $B$) to ensure that our learned $\hat{B}_n$'s can capture $\theta_n$'s for \emph{most} task $n$, in an average sense.
>
> - ``In Figure 1a) it looks like the Boss algorithm performs worse after a new dimension for the subspace is incremented at n= 2501, does that mean that the dimension of the subspace was falsely estimated?''
>
> $=>$ Our estimator $\hat{B}_n$ is always in $\mathbb{R}^{d \times m}$, so the subspace dimension estimate is always $m$. If the reviewer meant ``the newly revealed direction of the subspace was falsely estimated'', we agree.
>
> BOSS uses the estimated $\hat{B}$ to efficiently estimate $\hat{\theta}_n$. When a new dimension is revealed at n=2501, it takes a while for BOSS to have an accurate estimation $\hat{B}_n$ with respect to $B_n$, the subspace spanned by all $\{\theta_i: i=1,..,n\}$ shown so far. Fig 1b and 1c show that if the expert set covers the underlying $B$ (green plot), BOSS can adapt and learn very fast.
>
> - ``The presentation of the actual meta learning could be made clearer. The EWA algorithm for the meta learning procedure should not just be referenced but actually appear in the paper''
>
> $=>$ Thank you for your suggestion. We will include the full EWA algorithm in the appendix in the final version.

---

> > ### Comment · Reviewer_9RaC · 2024-08-13
> >
> > Thank you for your detailed answers to my questions and concerns. I will take them into consideration for the reviewers discussion period.

---

### Official Review · Reviewer_LLug · 2024-07-21

**Soundness:** 3
**Presentation:** 4
**Contribution:** 3
**Rating:** 7
**Confidence:** 4

**Summary:**

This paper addresses the problem of representation transfer in a sequential multi-task linear bandit problem. The main objective in the paper is to remove the task diversity assumption made in Qin et al (2022), which places a constraint on any subsequence of tasks observed. The paper proposes an algorithm that for each task performs exploration with a constant probability, and uses the exponential weighted average algorithm to choose the exploration probability over a large set of arm vectors. The paper proves an upper bound on the cumulative regret without making the task diversity assumption and demonstrate the performance in an experiment.

**Strengths:**

* The paper presents the results in an engaging and clear manner and the algorithms proposed are intuitive.
* Removal of the task diversity assumption appears to be significant advance.

**Weaknesses:**

* The problem setting emphasizes the sequential nature of the tasks. Specifically, the algorithm needs to collect all samples from $\theta_1$ task before moving on to $\theta_2$, i.e., the tasks cannot be interleaved. It would be helpful to outline some situations where this aspect of the problem setup is important.
* The algorithm needs to know the value of $m, N, \tau$ to choose the exploration probability. The proposed algorithm performs better than the "individual task baseline" only under a specific parameter regimes (see line 242).
* The theoretically valid size of the set of experts is extremely large for reasonable parameter values (lines 408 and 284). Is that one of the reasons why BOSS only performs better when we have very large number of tasks?
* There is a gap to the known lower bound, which is suggested to be due to the task diversity assumption not being met.

**Questions:**

* If possible, it would be nice to demonstrate the impact of the task diversity assumption. Specifically, could we construct a simulated environment where the assumption is not met by a significantly large fraction of subsequences, and the performance of BOSS is better than SeqRepL Qin et al (2022) ? Maybe your experiment already demonstrates that but I missed it. Could you clarify?

---

> ### Author Rebuttal · Authors · 2024-08-07
>
> - ``There is a gap to the known lower bound, which is suggested to be due to the task diversity assumption not being met.''
>
> **Before our work, no regret bounds that are $o(N d \sqrt{\tau})$ were known for Sequential representation transfer multi-task linear bandits without Task Diversity assumption**.
>
> In addition, we don't know if our upper bound can be improved. Even with task diversity, the upper and lower bounds of Qin et al. do not exactly match (see Table 1)
>
> - ``The theoretically valid size of the set of experts is extremely large for reasonable parameter values, is that one of the reasons why BOSS only performs better when we have very large number of tasks?''
>
> **Yes.** $|\mathcal{E}^\epsilon|$ directly affects the $\frac{\tau \ln|\mathcal{E}^\epsilon|}{p} = \tilde{O}( \frac{\tau d m } p )$ term in the regret bound, and will subsequently affect the $N^{2/3} \tau^{2/3} d m^{1/3}$ term in the regret bound.  If we can make this term smaller, we can broaden the parameter regimes when our regret bound is better than the $N d \sqrt{\tau}$ baseline. We are also working on a method that is more computationally efficient in the future.
>
> The reason why we need large $N$ is explained in the global answer above.
>
> - ``It would be nice to demonstrate the impact of the task diversity assumption''
>
> $=>$ Our experiments in the submission are designed without the Task Diversity assumption, as shown in the caption of Fig 1 (a new dimension of the underlying subspace $B$ is revealed at tasks 1, 2500, and 3500). We also add some extra experiment results to highlight how BOSS outperforms SeqRepL when the parameters length is larger ($0.8 \leq \|\theta_n\|_2 \leq 1$) in the attached PDF file of the global answer. Furthermore, we also add another set of experiments when the Task Diversity assumption is satisfied, where SeqRepL outperforms BOSS as expected.
>
> The first figure is the experiment results without the Task Diversity assumption and the second is with Task Diversity. Different from the main paper, we ensure that $0.8 \leq \|\theta_n\|_2 \leq 1$ highlights how badly SeqRepL performs when the Task Diversity assumption is violated. We also made some small changes to the hyper-parameters, as shown in the figures' captions.

---

> > ### Comment · Reviewer_LLug · 2024-08-14
> >
> > Thank you for the response, I have increased my score.

---

> ### Author Response · Authors · 2024-08-14
>
> As the author-reviewer discussion period is coming to a close, we kindly ask for your review of our response. This ensures that any additional questions or feedback you may have can be addressed before the discussion period ends. Thank you for your time!

---

### Author Rebuttal · Authors · 2024-08-07

We thank the reviewers for their insightful feedback.

In this work, we introduce the first provable representation transfer algorithm for sequential linear bandits without relying on the task diversity assumption. Unlike in parallel settings or sequential settings that assume task diversity, the learner now has to address the $\textit{unique challenges}$ in meta-exploration, which involves carefully deciding when to acquire more information on the low-dimensional representation.

We are encouraged that the reviewers found our work clear (R1, R2, R4), intuitive (R1, R4), and novel (R2, R3).
We are also glad that the reviewers recognized the significance of removing the task diversity assumption (R1, R2, R4) in exchange for some mild assumptions (R2) within the more challenging sequential setting (R4).

In addition, our theoretical guarantee and experiment are also appreciated (R1, R3, R4). We address some common questions below and the specific questions in each reviewer's section.

- ``The proposed algorithm performs better than the ”individual task baseline” only under a specific parameter regimes (R1, R4), $N \geq md$ and $d^2 \leq \tau \leq (N/m)^2$, which means the algorithm can only learn for a finite time (R3). [Does] BOSS only performs better [than the individual single-task baseline] when we have very large number of tasks? (R1)''

$ \textbf{We wish to point out that our work gives the first nontrivial result towards sequential multi-task linear bandit without task diversity assumption.}$
Before our work, no regret bounds better than individual single-task baseline (i.e. $o(N d \sqrt{\tau})$) were known for this setting.

Our main motivation for this paper was to show that sequential representation transfer is indeed possible without the task diversity assumption. Proving this opens the floodgates for developing more sample-efficient algorithms.

We can rephrase the parameter regime where our regret bound is better than the individual single-task baseline as:
$\tau \gg d^2$ and $N \gg m \sqrt{\tau}$. So, indeed, BOSS performs better when $N$ is large. As mentioned in the paper, we think that broadening the parameter regime for improvement is an important problem.

If we interpret R3's concern correctly, $\tau$ being capped at $(N/m)^2$ is a limitation of our paper. We agree -- this is due to the existence of the burn-in term $\tau m d$ (when $\tau > (N/m)^2$, this is greater than $N d \sqrt{\tau}$). Removing this burn-in term is an interesting open problem.

- ``The algorithm needs to know the value of $m, d, N, \tau$. Can this be avoided?'' (R1, R4)

$\textbf{Maybe}$. The requirement for knowledge of $m$ can be relaxed to knowing an upper bound of $m$. Removing this knowledge requires a change of approach, such as low-rank matrix optimization, as in Cella et al. 2023 or additional assumption, as in Bilaj et al. 2024. Cella et al. 2023 is in the parallel setting, which is not applicable here, and Bilaj et al. 2024's guarantee can be as large as $O(Nd\sqrt{\tau})$ as discussed in line 91.

The requirement for $\tau$ and $d$ is not too bad since the action space is given to us ahead of time, so $d$ is known, and $\tau$ can be known after finishing the first task.

Using an adaptive design to not be dependent on $N$ is likely to be possible with the doubling-trick. The details are in the next question.

- ``Can we use an adaptive design (R4)? Can $p$ gradually vanishes (R2)?''

$\textbf{Yes}$. We think achieving adaptivity in $N$ can be done using a doubling trick.

Specifically, in phase $i$, run our algorithm with the assumption that there are $2^i$ total tasks in this phase. The modified algorithm has a meta-regret guarantee that is within a constant of the algorithm that knows $N$. This implicitly gives an adaptive setting of meta-exploration probability $p$ that is decaying over time.

- ``Can our results translated to the interleaved (R1) or Parallel (R4) setting?''

$=>$ Our algorithm is designed to tackle the unique challenges in sequential transfer without the task diversity assumption, and so applying it to the parallel setting would result in losing the intended benefits.

We don't think that our result can be easily adapted to the parallel setting and achieve a better guarantee than Hu et al. 2021.

The interleaved setting can be seen as a unified model for both the Sequential and Parallel settings. Since these two are fundamentally different, investigating the interleaved setting is left for future work.

- ``The labels on the y-axis in the figures need to be fixed (R2, R3)''

$\textbf{Yes.}$ We will correct it as you suggest.

---

### Decision · Program_Chairs · 2024-09-25

**Decision:**

Accept (poster)

**Comment:**

This paper addresses the problem of representation transfer in a sequential multi-task linear bandit problem. The main objective in the paper is to remove the task diversity assumption made in Qin et al (2022), which places a constraint on any subsequence of tasks observed. The paper proposes an algorithm that for each task performs exploration with a constant probability, and uses the exponential weighted average algorithm to choose the exploration probability over a large set of arm vectors. The paper proves an upper bound on the cumulative regret without making the task diversity assumption and demonstrate the performance in an experiment.

All reviewers believe this paper is above the bar of acceptance. The AC agrees and recommends acceptance.